# Transcriptional reprogramming by mutated IRF4 in lymphoma

Nikolai Schleussner [1,2,3,29], Pierre Cauchy [4,5,6,7,29], Vedran Franke [8], Maciej Giefing [9,10], Oriol Fornes [11], Naveen Vankadari [12], Salam A. Assi [5], Mariantonia Costanza [1,2,3], Marc A. Weniger [13], Altuna Akalin [8], Ioannis Anagnostopoulos [14], Thomas Bukur [15], Marco G. Casarotto [16], Frederik Damm [2], Oliver Daumke [17], Benjamin Edginton-White [5], J. Christof M. Gebhardt [18], Michael Grau [19,20], Stephan Grunwald [17], Martin-Leo Hansmann [21,22], Sylvia Hartmann [23], Lionel Huber [4], Eva Kärgel [24], Simone Lusatis [1,2,3], Daniel Noerenberg [2], Nadine Obier [4,5], Ulrich Pannicke [25], Anja Fischer [26], Anja Reisser [18], Andreas Rosenwald [14], Klaus Schwarz [25,27], Srinivasan Sundararaj [16], Andre Weilemann [20], Wiebke Winkler [1,2,3], Wendan Xu [20], Georg Lenz [20], Klaus Rajewsky [28], Wyeth W. Wasserman [11], Peter N. Cockerill [5], Claus Scheidereit [24], Reiner Siebert [10,26], Ralf Küppers [7,13], Rudolf Grosschedl [4], Martin Janz [1,2,3], Constanze Bonifer [5] & Stephan Mathas [1,2,3,7] ✉

Disease-causing mutations in genes encoding transcription factors (TFs) can affect TF interactions with their cognate DNA-binding motifs. Whether and how TF mutations impact upon the binding to TF composite elements (CE) and the interaction with other TFs is unclear. Here, we report a distinct mechanism of TF alteration in human lymphomas with perturbed B cell identity, in particular classic Hodgkin lymphoma. It is caused by a recurrent somatic missense mutation c.295 T > C (p.Cys99Arg; p.C99R) targeting the center of the DNA-binding domain of Interferon Regulatory Factor 4 (IRF4), a key TF in immune cells. IRF4-C99R fundamentally alters IRF4 DNA-binding, with loss-of-binding to canonical IRF motifs and neomorphic gain-of-binding to canonical and non-canonical IRF CEs. IRF4-C99R thoroughly modifies IRF4 function by blocking IRF4-dependent plasma cell induction, and up-regulates disease-specific genes in a non-canonical Activator Protein-1 (AP-1)-IRF-CE (AICE)-dependent manner. Our data explain how a single mutation causes a complex switch of TF specificity and gene regulation and open the perspective to specifically block the neomorphic DNA-binding activities of a mutant TF.

Deregulated transcription factor (TF) activities are major contributors towards malignant transformation, as particularly exemplified by various hematopoietic malignancies. One inherent feature of disturbed TF activities is the deregulation of cellular processes such as lineage maintenance, differentiation, growth, and survival, thus promoting oncogenic transformation[1–3]. Mutations targeting TF DNA-binding motifs can affect TF:DNA interaction and/or TF functionality[4–6], but it is currently unclear whether such mutations can influence the interaction with other TFs and thus impact upon the nature of binding to TF Composite Elements (CEs). TF binding to DNA frequently involves the

formation of multimeric complexes binding to CEs which display much higher affinity-binding compared to any of the partners binding alone[7,8]. For example, the Activator Protein-1 (AP-1) TF family typically binds as JUN/FOS dimer to the palindromic sequence 5'-TGASTCA-3'. However, the affinity of AP-1 DNA-binding is greatly increased when its binding is enhanced by contacts with the Nuclear Factor of Activated T cells (NFAT) at the respective CE. This CE reduces the extent to which AP-1 sites must conform to the consensus required for AP-1 alone, but also renders these sites dependent upon additional $Ca^{2+}$ signaling[9,10].

TF binding to DNA is a complex process, with arginine (Arg; R)-residues playing an important role in protein-DNA recognition[11,12]. For example, a cluster of loss-of-function *TP53* mutations affects various R-residues central to TP53:DNA interaction[13]. Also, TF Interferon Regulatory Factor 4 (IRF4) contains an arginine at amino acid (AA) position 98 in the α3-helix of its DNA-binding domain (DBD), which is essential for its interaction with DNA[14].

The expression of the IRF family member IRF4 is largely restricted to immune cells, where it exerts key regulatory functions[15,16]. IRF4 not only plays important roles in the activation of both B- and T-lymphoid cells, but is also required for the generation of germinal center (GC) B cells and orchestrates terminal B-cell differentiation, i.e., the formation of plasma cells[17–19]. Apart from these functions during normal lymphopoiesis, IRF4-dependency is a characteristic of various hematopoietic malignancies, including multiple myeloma, diffuse large B-cell lymphoma (DLBCL) subtypes, or T-cell lymphoma entities[20–22]. The strength of binding of IRF4 to its cognate DNA-binding motifs depends on the one hand on its expression levels[23,24], but also on the interaction with other TFs binding to CEs, as shown for Erythroblast transformation-specific (ETS)-IRF (EICE)[25] or AP-1-IRF CEs (AICE)[26–28]. The extent and nature of these interactions define the specificity and strength of IRF4-directed transcriptional regulation in a given cell type[23,24]. High-level IRF4 expression is characteristic for the Hodgkin/Reed-Sternberg (HRS) tumor cells of classic Hodgkin lymphoma (cHL), a common human B-cell-derived malignancy[29]. However, HRS cells lack the IRF4-instructed terminal B-cell differentiation gene expression program, including plasma cell genes[30,31], and instead up-regulate genes characteristic of other cellular lineages[30,32,33]. Furthermore, HRS cells are surrounded by an inflammatory cellular infiltrate attracted by abundantly produced cytokines, chemokines and cell surface receptors[30]. Only very few genetic events driving these features of Hodgkin lymphoma cells are known.

Here, we describe a distinct mechanism of TF alteration in human lymphomas, particularly cHL, involving a recurrent somatic missense mutation c.295 T > C (p.Cys99Arg; p.C99R) that targets the center of the DNA-binding domain of IRF4. We show that IRF4-C99R results in fundamental changes in IRF4's DNA-binding properties, combining loss-of-binding to canonical IRF motifs and neomorphic gain-of-binding to canonical and non-canonical IRF CEs. Functionally, we demonstrate that IRF4-C99R blocks IRF4-dependent plasma cell induction and up-regulates disease-specific genes in a non-canonical Activator Protein-1 (AP-1)-IRF-CE (AICE)-dependent manner. Our data explain how a single mutation causes a complex switch of TF specificity and gene regulation.

## Results

### The IRF4-C99R mutation is recurrent in human lymphoma
By mining and integrating both our own and additional published genomic and transcriptional data from well-characterized cHL cell lines, we identified and verified the same c.295 T > C (chr6:394,899 T > C; hg38) variant in the *IRF4* gene in 2 of 7 HL cell lines, namely the B-cell-derived HRS cell lines L428 and U-HO1 (Fig. 1a and Supplementary Fig. 1a). Based on various in silico analyses integrated in ANNOVAR (including SIFT, Polyphen2, MutationTaster, FATHMM, CADD score), this variant was uniformly predicted to be deleterious (Supplementary Table 1) and is completely absent in germline genomic databases

(gnomAD, accessed 2022/06/16). Furthermore, no germline non-synonymous single nucleotide variants were collated in gnomAD affecting the neighboring AAs 90–104 with the exception of a singleton allele (1/251478) carrying a missense mutation in AA100. In the HL cell lines, the c.295 T > C mutant allele was accompanied by at least one wild-type (WT) copy of the *IRF4* gene, and both WT and mutant *IRF4* mRNA transcripts were equally detected (Fig. 1a and Supplementary Fig. 1a). Since HRS cells are rare in the affected lymph nodes, we validated the presence of *IRF4* c.295 T > C in 4 of 20 primary cHL samples representing 3 of 19 cases (16%) by DNA-PCR of laser-microdissected HRS cells (Supplementary Table 2). The S104T mutation identified in L428 cells (Supplementary Fig. 1a) was not found in the primary cases, and thus not considered as recurrent. *IRF4* c.295 T > C has recently been described in Primary Mediastinal B Cell Lymphoma (PMBCL)[34], a lymphoma entity that shares distinct biological features with cHL. Parallel mining of targeted gene panel sequencing data from an unrelated large cohort of 486 PMBCL cases identified the same *IRF4* c.295 T > C mutation in 29 of the 486 cases (5.9%) (Supplementary Fig. 1b). In contrast, *IRF4* c.295 T > C is only rarely documented in other lymphoma types such as DLBCL (Supplementary Fig. 1b; refs. 35–37). Furthermore, the genomic location of the C99R c.295 T > C (chr6:394,899 T > C; hg38) mutation is within exon 3, and thus located >3 kb downstream of the transcription initiation site (TIS). In addition, it lacks the typical hotspot RGYW motif, indicating that this mutation is not caused by aberrant somatic hypermutation in B-cell lymphoma, which usually affects regions spanning about 2–2.5 kb downstream from the TIS[38,39].

IRF4 governs the plasma cell gene expression program at the stage of terminal B-cell differentiation[40], which largely lacks in HRS cells[31] despite high-level IRF4 expression across all subtypes (Supplementary Fig. 1c, d). In the *IRF4* c.295 T > C mutation, the basic AA arginine replaces the neutral AA cysteine (Cys; C) (p.Cys99Arg; C99R) at position AA 99, which is highly conserved in IRF4 from humans to zebrafish and also within the DBD of most other IRF family proteins (Fig. 1a and Supplementary Fig. 1e). C99R is located in the center of the α3-recognition helix of the DBD of IRF4 and is positioned immediately adjacent to Arg98, which is essential for specific IRF4 DNA-binding[14]. This finding suggested that C99R might interfere with the formation of IRF4:DNA complexes and thus with IRF4's transcriptional activity.

### IRF4-C99R shows loss-of-function at ISRE but is functionally active
To characterize IRF4-C99R, we first explored its DNA-binding activity to the Interferon-Stimulated Response Element (ISRE) containing three consensus motifs 5'-GAAA-3' (Fig. 1b and Supplementary Fig. 1f), one of the key motifs recognized by IRFs[41,42]. Unlike IRF4-WT, IRF4-C99R did not bind to the ISRE at all, as demonstrated by Electrophoretic Mobility Shift Assay (EMSA). However, the recurrent nature of IRF4-C99R mutation and high-level expression in cHL suggested that this mutation may not merely constitute a loss-of-function aberration, but could possess additional, de novo functions. To analyze IRF4-C99R functionality, we generated tetracycline (Tet)-inducible IRF4-C99R and IRF4-WT expressing bulk cultures of BJAB B-cell non-Hodgkin lymphoma cells, which express endogenous IRF4 only at a low level (Supplementary Fig. 2a). Time course gene expression analyses revealed that IRF4-C99R altered the expression of a distinct, albeit fewer set of genes compared to IRF4-WT (Fig. 1c, Supplementary Fig. 2b–e, and Supplementary Data 1). Notably, IRF4-C99R was unable to induce plasma cell-specific genes (Supplementary Fig. 2f), in agreement with its lost ability to bind the canonical ISRE motif. IRF4-C99R rescued HRS cells as efficiently as IRF4-WT from cell death induced by small-hairpin RNA (shRNA)-mediated knock-down of endogenous IRF4 (Supplementary Fig. 2g) thus corroborating its functionality.

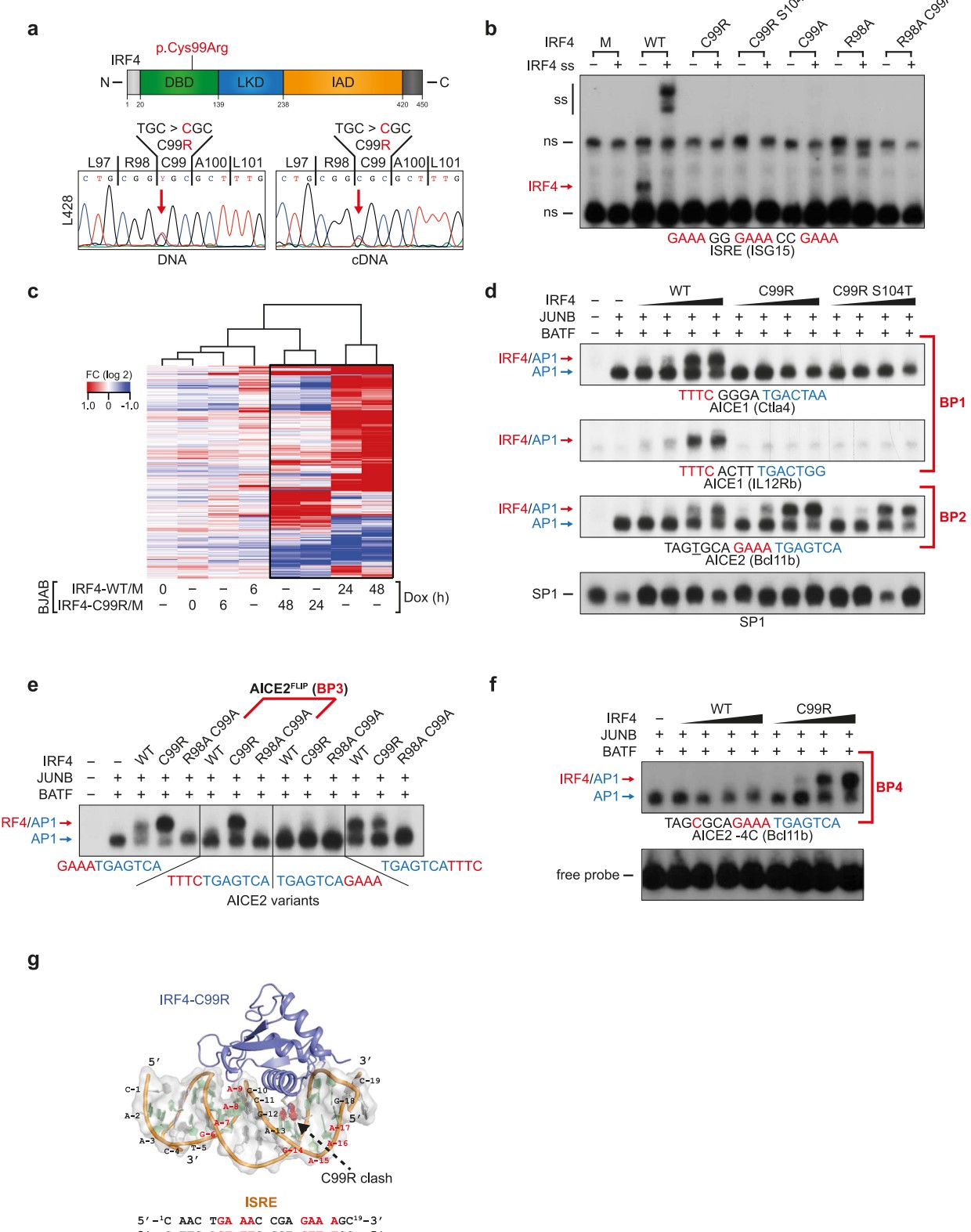

## IRF4-C99R fundamentally modifies IRF4's DNA-binding specificity

In contrast to the formation of low-affinity homodimer or multimeric complexes on ISRE DNA motifs, efficient IRF4 DNA-binding requires distinct partners such as ETS and AP-1 proteins at CEs[25–27]. Given the broad absence of ETS TFs in cHL[43,44], we considered the binding of IRF4 to EICE in HRS cells as being unlikely. However, constitutive AP-1 activity with high-level JUNB and BATF expression is a hallmark of HRS cells[45,46]. We therefore speculated that IRF4-C99R regulates gene expression by DNA-binding to the recently identified AICEs, either 5′-IRF(TTTC)/nnnn/AP-1(TGASTCA)-3′ with a spacing of 4 bp (AICE1) or 5′-IRF(GAAA)/AP-1(TGASTCA)-3′ with no spacing (AICE2)[26,47], which both regulate key transcriptional programs in immune cells[16]. To evaluate this hypothesis, we monitored the formation of IRF4-JUNB/BATF-DNA

**Fig. 1 | Characterization of IRF4-C99R functionality and fundamental DNA-binding alterations. a** Top, scheme of IRF4 with indication of p.Cys99Arg within the DBD. Bottom, *IRF4* Sanger sequencing reads of L428 DNA (left) and cDNA (right). DBD DNA-binding domain, LD linker domain, IAD IRF association domain. **b** EMSA using ISRE probe and nuclear extracts of HEK293 cells transfected with IRF4-WT or mutants thereof. M, Mock control; C99A, R98A, and R98AC99A, loss-of-function mutants. Red arrow, IRF4–DNA complex. ss, supershift. ns nonspecific. Data are representative of at least three independent experiments. **c** BJAB cells were analyzed for gene expression changes following tet-induction of the respective IRF4 variants for the indicated times. Hierarchical clustering of 348 differentially expressed genes that change expression in either IRF4-WT or IRF4-C99R vs. Mock (M) cells. Dox doxycycline. Log2 fold changes (FC) of at least twofold are indicated in the heatmap. Experiments were performed in biologically independent duplicates and log2 FC values are from two averaged replicates per condition. **d** Nuclear extracts of HEK293 cells transfected with JUNB and BATF and increasing amounts of IRF4 variants were analyzed by EMSA for binding at AICE1 (AICE1 (Ctla4) and AICE1 (IL12Rb)) and AICE2 (AICE (Bcl11b)) probes. BP binding pattern. Red and blue

arrows mark positions of IRF4-JUNB/BATF-DNA and JUNB/BATF-DNA complexes, respectively. EMSA of SP1 is shown as a control. Data are representative of at least three independent experiments. **e** EMSAs using wild-type AICE2 (BCL11b) probe (left) or variants thereof with reverse complement IRF motif (center left, AICE2^FLIP), or IRF motifs positioned 3′ relative the AP-1 motif (center right and right). Extracts and complex positions are as in (**d**). Data are representative of at least three independent experiments. **f** Binding of JUNB/BATF together with increasing amounts of IRF4-WT or IRF4-C99R to AICE2 (Bcl11b) with a T > C mutant thereof at position -4 relative to the IRF motif. Extracts and complex positions are as in (**d**). Data are representative of at least three independent experiments. **g** IRF4-C99R docking to ISRE. This modeling is based on the previous crystal structure of IRF4-WT:ISRE DNA (PDB: 7JM4), in which C99 was substituted with R. ISRE-sequence is shown underneath; numbered bases of the upper strand shown in the image above. Blue/slate, IRF4 (both WT and C99R); orange, DNA phosphate backbone; pale green, dA and dT; gray, dG and dC. Source data for (**b**, **d**–**f**) are provided in the Source Data file.

complexes at strong (labeled as "AICE1 (Ctla4)"), weak (AICE1 (IL12Rb)) or intermediate (AICE2 (Bcl11b)) affinity AICE motifs[26] (Fig. 1d and Supplementary Fig. 3a–c). While we observed a complete loss of IRF4-C99R binding at AICE1 (Ctla4) and AICE1 (IL12Rb) (designated as AICE Binding Pattern 1 (BP1)), IRF4-C99R-JUNB/BATF binding at AICE2 (Bcl11b) was enhanced compared to IRF4-WT (BP2) (Fig. 1d). IRF4-C99RS104T behaved similar to IRF4-C99R (Fig. 1b, d and Supplementary Fig. 3a), and, as it is not a recurrent mutation, it was not included in further experiments. Strikingly, reverse complementing the IRF motif in AICE2 (Bcl11b) from 5′-GAAA-3′ to 5′-TTTC-3′ (referred to as AICE2^FLIP) revealed formation of mutant IRF4-C99R-JUNB/BATF-DNA complexes only (Fig. 1e, AICE2^FLIP, BP3; Supplementary Fig. 3d). Moreover, formation of AICE complexes usually requires a thymine located at −4 bp (-4T) relative to AICE2 (referred to as AICE2^-4T)[47]. IRF4-C99R overrides this restriction, as it forms strong DNA-binding complexes in the absence of -4T, which causes loss of IRF4-WT binding (Fig. 1f; AICE2^-4C; BP4). Similarly, altered binding patterns of IRF4-C99R compared to IRF4-WT were observed together with c-JUN (JUN)/BATF heterodimeric AP-1 complexes (Supplementary Fig. 3e, f).

Furthermore, we performed structural modeling analysis to provide additional information on how the IRF4-C99R mutation influences the interaction with the ISRE and AICE1 DNA-binding motifs (Fig. 1g, Supplementary Fig. 4, and Supplementary Table 3). For the structural models, the initial structures of IRF4 and DNA were obtained from our previous crystal structure (PDB:7JM4), and the most viable models were considered based on resultant docking parameters such as HADDOCK score, cluster size and desolvation energy. As shown in Fig. 1g for the IRF4:ISRE interaction, a direct replacement of C99 with Arg resulted in a steric clash with DNA bases. To accommodate the interaction between C99R and ISRE, the dsDNA had to either bend and/or kink as shown in the overlay model of WT/C99R with ISRE (Supplementary Fig. 4a). For IRF4-C99R:AICE1 binding, the poor docking scores (Supplementary Table 3) indicated that it was highly unlikely that a significant interaction could take place as reflected by the poor energy-minimized structure (Supplementary Fig. 4b). Thus, even though our AICE1-modeling might be limited due to DNA distortions, it suggests that IRF4-C99R does not bind to AICE1, which is consistent with our EMSA results.

These patterns of alterations of the IRF4-C99R DNA-binding properties were also observed with recombinant proteins comprising just the DBDs of IRF4 (AA 20–139), JUNB (AA 269–329), and BATF (AA 28–87) only (Supplementary Fig. 5a–c). In addition, we visualized the DNA-bound fraction of IRF4-C99R or IRF4-WT by single-molecule fluorescence microscopy and interlaced time-lapse illumination[48], which revealed comparable percentages of long-bound DNA contacts (>2 s) of IRF4-C99R compared to IRF4-WT molecules (Supplementary Fig. 5d). Together, our data demonstrate a unique combined loss-and-

gain of DNA-binding preferences by IRF4-C99R, and, in particular, neomorphic binding activity to AICE2-like motifs.

## Globally altered IRF4 DNA-binding patterns and cooperative activities in IRF4-C99R lymphoma cells

We next aimed to obtain global data supporting the above findings by interrogating our HL cell line models. To specifically map accessible chromatin in HRS cells in detail, we first generated high-resolution genome-wide DNaseI hypersensitive site (DHS) and digital footprinting data from the HRS cell lines L428, harbouring IRF4-C99R, and KM-H2, expressing IRF4-WT, as well as the non-Hodgkin, non-IRF4 expressing REH cells as a control (Supplementary Fig. 6a). The analyses of DNaseI cutting frequencies revealed protection against DNaseI digestion, indicative of occupancy by protein complexes, together with elevated accessibility of the flanking regions at AICE2 (BP2), AICE2^FLIP (BP3), AICE2^-4T and AICE2^4C (BP4) only in HRS cells (Fig. 2a). Notably, and in line with our DNA-binding experiments (see Fig. 1), these motifs were highest enriched and protected in L428^IRF4-C99R (Fig. 2a). Co-localization analysis of these AICE2 motifs in L428^IRF4-C99R cells revealed a specific cluster corresponding to mutant-specific sites co-localizing with AP-1 motifs but not with those of other TFs typically involved in B and HL cell gene regulation (Supplementary Fig. 6b, left), which was not observed in KM-H2^IRF4-WT cells (Supplementary Fig. 6b, right). These findings again supported the idea of IRF4-C99R conferring cells with divergent expression profiles.

To define groups of L428 or KM-H2-specific DHSs, we determined the ratio of tag counts between L428^IRF4-C99R and KM-H2^IRF4-WT cells and ranked them according to their fold change in DNaseI-seq signal (Fig. 2b; groups 1–3). The L428^IRF4-C99R-specific DHSs (i.e., group 3) correlated with upregulated gene expression in these cells (Fig. 2b). We then determined the enrichments of AICE2 (BP2), AICE2^FLIP (BP3), AP-1 and ISRE motifs in the different DHS groups. L428^IRF4-C99R-specific DHSs were enriched for AICE2, AICE2^FLIP and AP-1 motifs, but depleted for ISRE motifs, whereas KM-H2^IRF4-WT-specific DHSs were depleted for AICE2, AICE2^FLIP and AP-1 motifs, but enriched for ISRE (Fig. 2c). An unbiased search for TF motifs in the cell line-specific DHSs using HOMER revealed AICE2 and AICE2^FLIP as 2 of the most highly enriched motifs in L428^IRF4-C99R-specific DHSs but not in KM-H2^IRF4-WT-specific DHSs sites (Fig. 2d). Conversely, ISRE motifs were enriched in KM-H2^IRF4-WT- but not L428^IRF4-C99R-specific DHSs, again suggesting that IRF4-C99R shifts binding to distinct AICE2 motifs (Fig. 2d). Parallel DHS analyses comparing L428 versus REH (Supplementary Fig. 7a, b) or versus publicly available DHS data from lymphoblastoid GM12878 B cells (Supplementary Fig. 7c, d) also revealed specific enrichment of AICE2 motifs in L428^IRF4-C99R-specific DHSs. Gene set enrichment analysis (GSEA) in DHSs from L428^IRF4-C99R versus KM-H2^IRF4-WT cells revealed an increased presence of footprinted AICE2 motif in upregulated

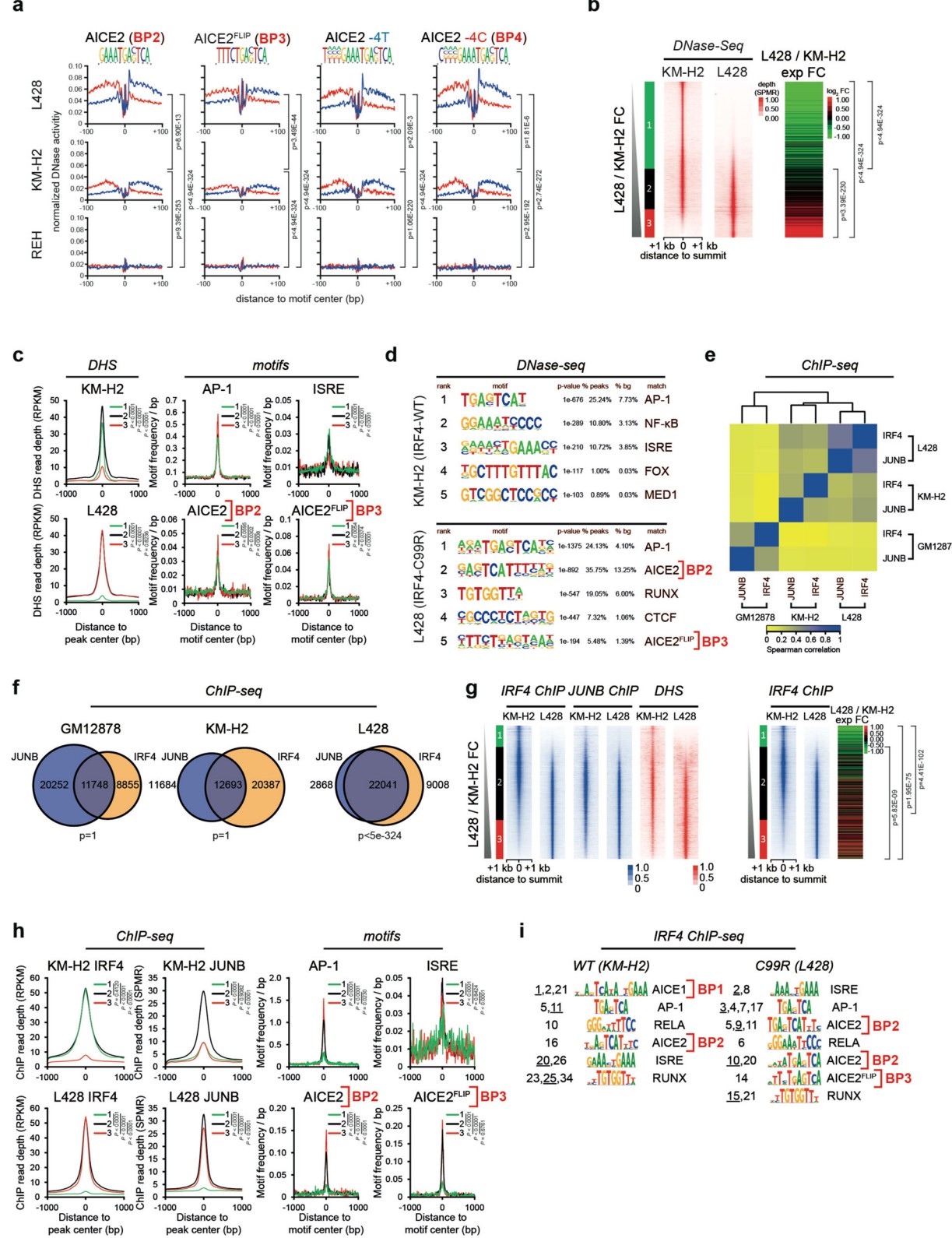

genes (Supplementary Fig. 7e), arguing for the functional relevance of this motif in AICE2 L428$^{IRF4-C99R}$ cells.

A drawback of tools like HOMER is, that, although they are excellent at identifying global consensus binding motifs, they have difficulties in identifying different and slightly degenerate versions of the same motif, as found in CEs. In addition, these algorithms focus on the core motifs while ignoring the flanking nucleotides. To overcome

these limitations, we used a novel deep-learning tool, ExplaiNN (explainable neural networks)[49], to separately discover motifs de novo in the cell line-specific DHS datasets. This analysis confirmed that AICE1 (BP1) in KM-H2$^{IRF4-WT}$ and AICE2 (BP2) in L428$^{IRF4-C99R}$ were among the most important motifs (Supplementary Fig. 8a–c).

Finally, we performed genome-wide JUNB and IRF4 Chromatin Immunoprecipitation (ChIP)-Seq analyses in L428$^{IRF4-C99R}$ and

**Fig. 2 | IRF4-C99R is associated with genome-wide increased and distinct DNA-binding patterns at canonical and non-canonical AICE2 sites in C99R mutation-positive lymphoma cells. a** Digital genomic footprinting (DGF) analyses showing occupancy at AICE2, AICE2$^{FLIP}$, AICE2$^{-4T}$, and AICE2$^{-4C}$ sites (horizontal) in L428$^{IRF4-C99R}$ HL cells, KM-H2$^{IRF4-WT}$ HL and REH non-Hodgkin cells (vertical). Red and blue lines represent the forward and reverse strands. **b** Correlation between L428$^{IRF4-C99R}$/KM-H2$^{IRF4-C99R}$ DNaseI-Seq fold change defining three classes of elements, designated groups 1–3, (left) and log2 gene expression fold change (right). **c** Motif frequencies in DHSs defined in (**b**). **d** HOMER de novo motif discovery in specific DHSs defined in (**b**). **e** Heatmap showing Spearman's correlation clustering from IRF4 and JUNB ChIP-Seq experiments on the union of IRF4 peaks from L428$^{IRF4-C99R}$, KM-H2$^{IRF4-WT}$ and GM12878 cells. **f** Venn diagram-overlaps between IRF4 and JUNB ChIP peaks in GM12878, KM-H2$^{IRF4-WT}$ and L428$^{IRF4-C99R}$ cells. **g** L428/KM-H2 IRF4 ChIP peak fold change analyses (left) showing corresponding JUNB ChIP peaks (center), DHSs (right) as well as gene expression fold changes (far right). **h** Motif frequencies in KM-H2$^{IRF4-WT}$-, shared and L428$^{IRF4-C99R}$-specific ChIP peaks. **i** De novo motif discovery in specific ChIP-seq datasets using ExplaiNN. Motifs are ranked by their importance (left). When more than one motif of the same class was identified, the rank of the displayed motif is underlined. Only motifs that could be annotated with a biological representation are shown. All DGF/DNase-Seq/ChIP-Seq/RNA-Seq values are from two averaged replicates per condition. *P* values were determined by two-tailed unpaired *t* test without adjustment for multiple comparisons.

KM-H2$^{IRF4-WT}$ cells (Fig. 2e−i and Supplementary Fig. 9a). We included publicly available IRF4 and JUNB ChIP-Seq data from GM12878 cells, since both IRF4 and JUNB are virtually not expressed in REH cells. Sequences within IRF4-JUNB ChIP peaks clustered closely together (Fig. 2e) and showed a greater overlap (Fig. 2f) in L428$^{IRF4-C99R}$ cells (Dice score: 0.7877) compared to KM-H2$^{IRF4-WT}$ and GM12878 cells (Dice scores: 0.4418 and 0.4478, respectively), in line with enforced binding of IRF4-C99R to IRF and AP-1 CEs. Although IRF4 ChIP peak frequency was higher in both HRS cell lines compared to GM12878 (Fig. 2f), the overlap with JUNB was much lower in KM-H2$^{IRF4-WT}$ cells. When individually ranked, IRF4 and JUNB showed highly similar binding patterns in L428$^{IRF4-C99R}$ but not in KM-H2$^{IRF4-WT}$ cells, corresponded to open chromatin regions and were associated with increased gene expression (Fig. 2g). Consistent with these analyses and motif discovery results from DHS datasets, de novo motif analyses by HOMER (Supplementary Fig. 9b) and supervised motif injection (Fig. 2h) showed increased frequencies of AICE2 (BP2) and AICE2$^{FLIP}$ (BP3) motifs in L428$^{IRF4-C99R}$-specific IRF4 ChIP peaks, while conversely showing lower ISRE motif frequencies, when compared to KM-H2$^{IRF4-WT}$ specific ChIP peaks. These findings were also observed when IRF4 and JUNB chromatin binding patterns of L428 were compared against GM12878 cells (Supplementary Fig. 9c, d). Importantly, GSEA revealed that IRF4 and JUNB ChIP peaks were associated with increased gene expression in L428$^{IRF4-C99R}$ but not KM-H2$^{IRF4-WT}$ cells (Supplementary Fig. 9e).

Again, we performed de novo motif discovery using ExplaiNN, but this time in the ChIP-seq datasets, and found that AICE1 (BP1) was the most important motif in KM-H2$^{IRF4-WT}$ cells, but was not identified in L428$^{IRF4-C99R}$ cells (Fig. 2i and Supplementary Figs. 8a and 10). AICE2 (BP2) emerged among the most important motifs in both datasets, with more importance in L428$^{IRF4-C99R}$ cells, wherein a total of five motif types (vs one in KM-H2$^{IRF4-WT}$) were identified (Fig. 2i and Supplementary Fig. 10). The analyses also revealed the unique importance of AICE2$^{FLIP}$ (BP3) in L428$^{IRF4-C99R}$ cells. These results agree with our DNA-binding studies (Fig. 1), and further support the notion that IRF4-C99R fundamentally alters IRF4 genome-wide DNA-binding patterns in lymphoma cells and enforces cooperative binding with AP-1/JUN TFs at distinct neo-AICEs.

## IRF4-C99R disrupts IRF4 function and reprograms gene expression in primary B cells

To further explore the functional consequences of IRF4-C99R expression in B cells, we retrovirally transduced primary mouse C57BL/6 splenic B cells with IRF4-WT, IRF4-C99R, or the loss-of-function (LOF) variant IRF4-R98AC99A as a control (Fig. 3a and Supplementary Fig. 11a). Within the known IRF4-R98AC99A LOF variant, the residues R98 and C99, which are critically involved in the formation of IRF4:DNA complexes, were both replaced by alanin (A) abolishing IRF(4) DNA-binding and function[14,50–52]. Culturing of B cells with LPS and IL−4 led to robust endogenous IRF4 expression (Supplementary Fig. 11a) and resulted in induction of around 30% plasmablasts, characterized by a CD138$^{high}$ and B220$^{low}$ phenotype (Fig. 3a). The same result was obtained after ectopic expression of the non-functional

IRF4-R98AC99A variant (Fig. 3a). Following ectopic expression of IRF4-WT, ~70% of the cells converted to a plasmablast phenotype. In contrast, IRF4-C99R reduced the number of developing plasmablasts, i.e., blocked inherent plasmablast formation, arguing for a dominant-negative function of IRF4-C99R with respect to terminal B-cell differentiation (Fig. 3a). To examine alterations in gene expression, we isolated mouse C57BL/6 splenic B cells transduced with the different IRF4 variants followed by RNA-seq analyses (Fig. 3b−f and Supplementary Data 2). Overall, the data from the respective transfectants clustered separately, with IRF4-C99R showing a transcriptional profile more similar to the R98AC99A LOF variant than to IRF4-WT (Fig. 3b). IRF4-C99R regulated a reduced set of genes (Fig. 3c), encompassing a broad loss of IRF4-WT target gene expression along with a gain of novel targets (Fig. 3d, Supplementary Fig. 11b, and Supplementary Data 2). Integration of the mRNA expression profiles of the modified splenic B cells with those from various hematopoietic cell types showed an IRF4-C99R-regulated block of overall IRF4-WT-induced and plasma cell-specific gene expression (Fig. 3e), confirming that IRF4-C99R is unable to instruct the IRF4-directed plasma cell program. Concomitantly, IRF4-C99R upregulated myeloid-associated genes (Fig. 3e, f), phenocopying a central feature of cHL tumor cells[30,32]. Together, these data confirmed the fundamental changes in IRF4-C99R-dependent gene regulation and function compared to IRF4-WT.

## IRF4-C99R activates lymphoma-specific gene expression *via* non-canonical AICEs

To directly link IRF4-C99R-regulated genes to those specifically inherent to HRS cells of cHL, we integrated our RNA-seq data from the splenic B cells with HRS cell-specific gene expression profiles (Fig. 4a and Supplementary Fig. 12a). The latter were deduced from published microarray data as well as mRNA-seq-based gene expression profiles of cHL and non-Hodgkin lymphoma cells. Among the most HRS cell-specifically expressed genes which were upregulated exclusively by IRF4-C99R, but not by IRF4-WT, were *GATA3*, *CCL5* (also called *RANTES*), and *TNFRSF8* (*CD30*), all three being among the most prominent cHL hallmark genes[31,53], together with *CD80*, *PDE4D*, and *CASP6* (Fig. 4a and Supplementary Fig. 12a).

To further dissect the mechanism of the IRF4-C99R-specific induction of these genes, we reanalyzed our ChIP-Seq-data for IRF4-JUNB ChIP peaks specific to L428$^{IRF4-C99R}$ cells, but not found in KM-H2$^{IRF4-WT}$ cells. Focusing on regions regulating *GATA3* expression, we identified several AICE2-like CEs among the L428$^{IRF4-C99R}$-specific IRF4-JUNB ChIP peaks, designated as *GATA3*Peak_1 (5′-TGAGTCA$\underline{GAGA}$-3′; the IRF-part of the binding motif is underlined), *GATA3*Peak_2 (5′-TAAATGAGTCA-3′) and *GATA3*Peak_3 (5′-$\underline{GGAA}$TGAGTCA-3′) (Fig. 4b, left). DNA-binding studies demonstrated that IRF4-C99R forms IRF-AP-1 composite complexes at these sites, whereas IRF4-WT did not bind to these sequences (Fig. 4b, right, AP-1 consisting of JUNB/BATF heterodimers; Supplementary Fig. 12b, c, AP-1 consisting of JUNB/BATF3 heterodimers). Of note, none of these sites contained canonical 5′-GAAA-3′ IRF motifs, but instead noncanonical degenerate variants thereof. These results pointed to the increased flexibility of IRF4 R99 compared to C99 at these motifs, similar to our observation made for

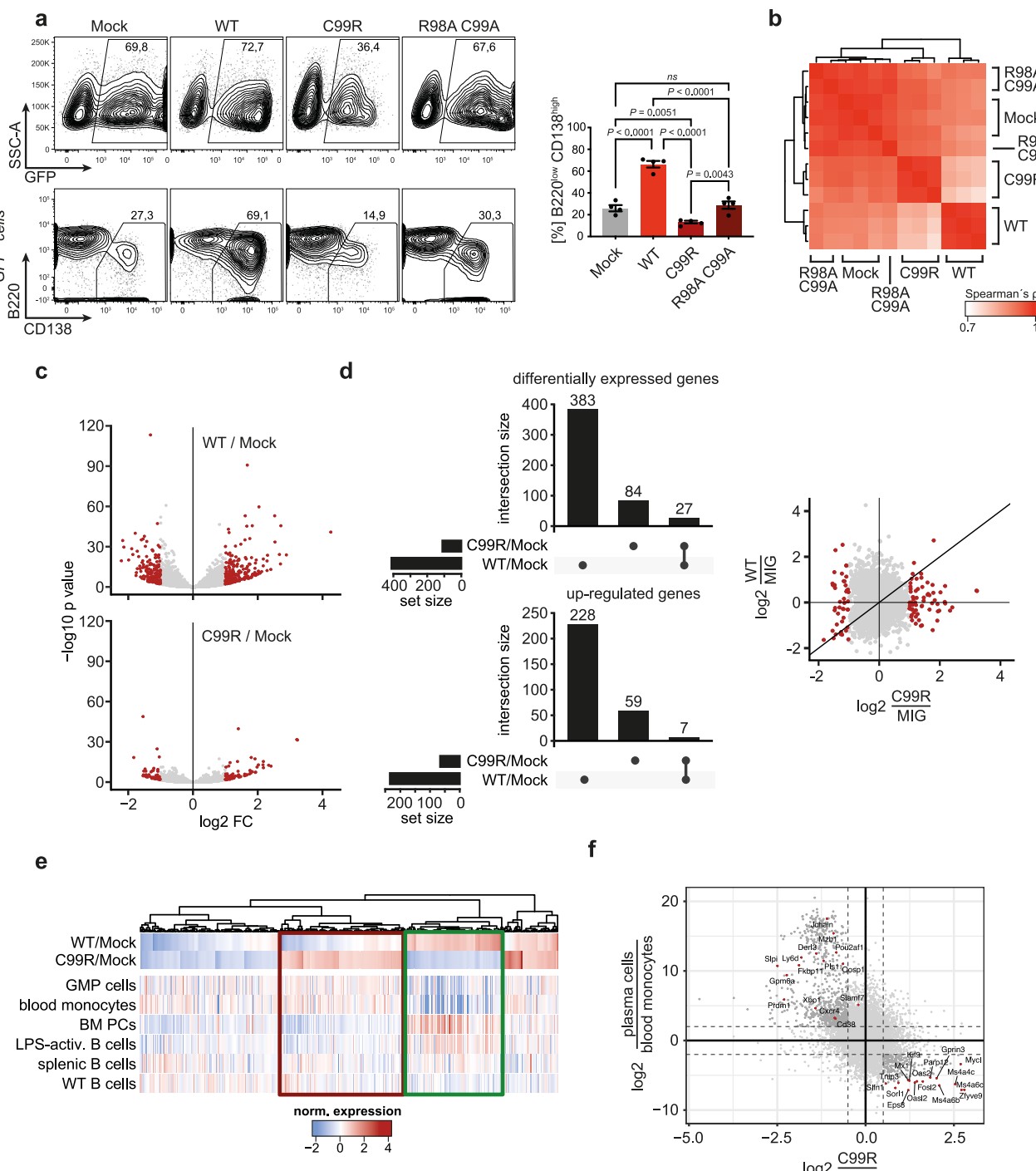

**Fig. 3 | IRF4-C99R blocks IRF4-dependent plasma cell induction and regulates less but distinct genes compared to IRF4-WT. a** Following culture with LPS + IL-4, C57BL/6 mouse splenic B cells were transduced with MIG control retrovirus (MIG-ctrl; Mock), IRF4-WT, IRF4-C99R or, as a further control, IRF4-R98AC99A. Transduced GFP[+] cells were analyzed by flow cytometry for expression of CD138 and B220. Top left panels, the indication of the percentage of living transduced cells in representative FACS profiles. Bottom left panels, analysis of CD138 and B220 in gated GFP[+] cells. The percentage of CD138[high]B220[low] cells is indicated. Right, the mean ± SEM of $n = 4$ independent experiments is shown. $P$ values were determined by two-tailed unpaired Student's $t$ test; ns, not significant. **b–f** Isolated murine splenic B cells transduced with IRF4-WT, IRF4-C99R, IRF4-R98AC99A, or MIG control (Mock) were analyzed by RNA-Seq. Experiments were performed in biologically independent triplicates ($n = 3$). **b** Spearman correlation of the various samples. Note, that the Mock and the IRF4-R98AC99A-LOF transduced cells cluster together, and that IRF4-C99R clusters in between these and IRF4-WT samples. **c** Volcano plots of genes differentially regulated between IRF4-WT versus Mock and

IRF4-C99R versus Mock (MIG-ctrl). Note, that IRF4-C99R regulates less genes compared to IRF4-WT. **d** IRF4-C99R and IRF-WT regulated genes show only low overlap, as shown in UpSet plots for overall differentially regulated genes (left, top panel) and upregulated genes (left, bottom panel), as well as in overall comparisons of complete transcriptomes (right panel). **e** Differentially regulated genes by IRF4-C99R were compared to gene expression of lymphoid and myeloid cell types. Note, that the IRF4-C99R-downregulated genes correspond to genes expressed in plasma cells or IRF4-WT-induced genes (green rectangle), whereas the IRF4-C99R-upregulated genes show specific expression in myeloid cells (red rectangle). GMP granulocyte/monocyte progenitor, BM bone marrow, PC plasma cell. **f** Comparison of IRF4-C99R fold change with ratio of gene expression from plasma cells and monocytes. The Spearman correlation between the log2 fold changes of plasma cells/blood monocytes versus C99R/WT was −0.39, with a $P$ value less than $10^{-5}$. The number of genes compared was 12642. Source data for figure part 3a, right are provided in the Source Data file.

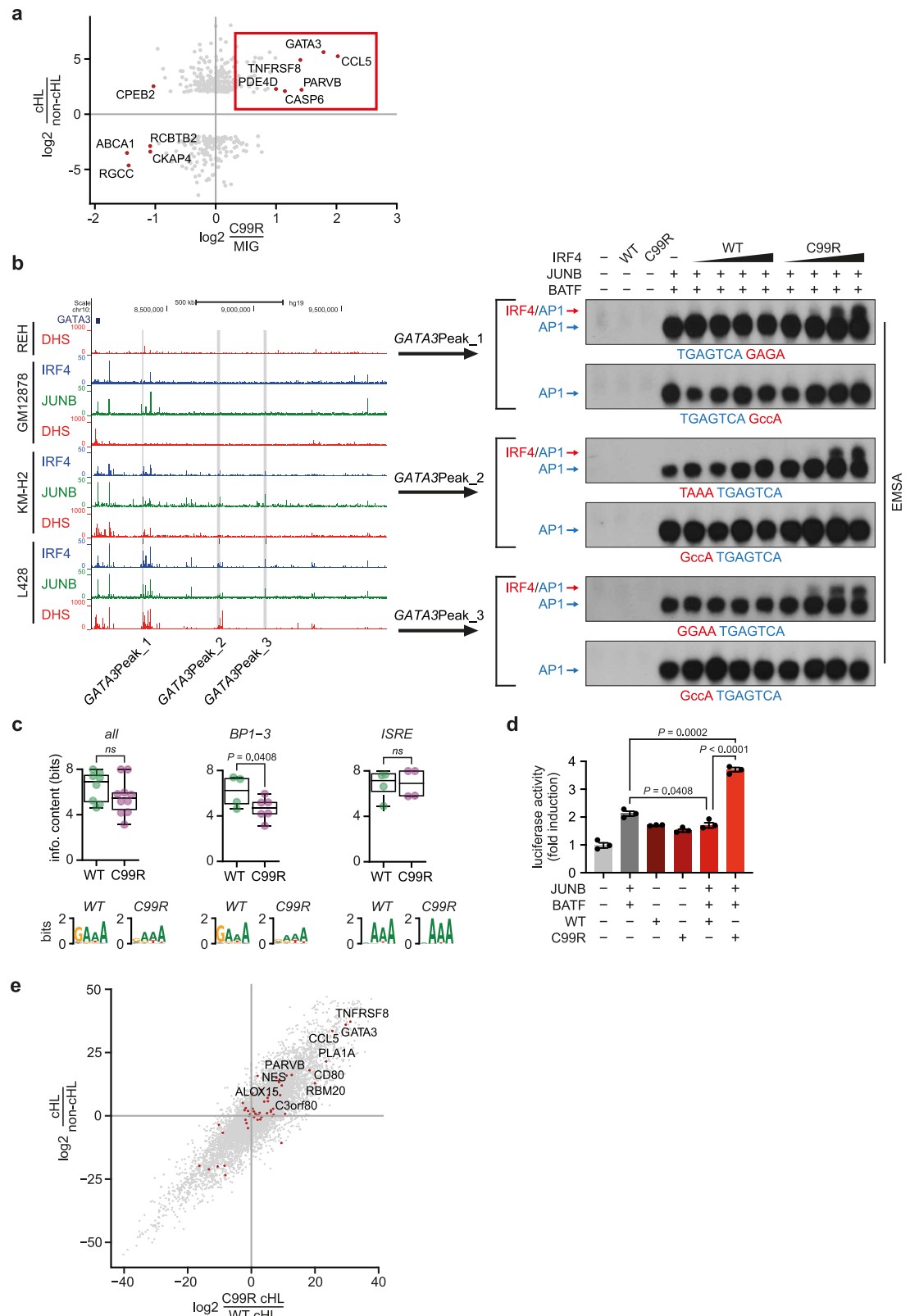

the AICE2 variants described in Fig. 1e–g. The increased binding capacity of IRF4-C99R to degenerate half-ISRE containing motifs was mirrored in the observation that IRF-containing motifs identified in the ChIP-seq data using ExplaiNN were more degenerated in L428[IRF4-C99R] cells compared to KM-H2[IRF4-WT] cells (Fig. 4c), which was most pronounced for AICE motifs (Fig. 4c, center). We confirmed IRF4-C99R-mediated transcriptional activity of the *GATA3*Peak_1 element by the

analysis of luciferase reporter constructs (Fig. 4d and Supplementary Fig. 12d). Here, IRF4-C99R specifically enhanced the luciferase activity in combination with the AP-1 TFs JUNB and BATF, whereas IRF4-WT did not. Finally, the comparison of expression profiles of HRS cell lines harboring an IRF4-C99R mutation with those lacking IRF4-C99R in relationship with cHL-specific genes showed that the cHL hallmark genes *GATA3*, *CCL5* and *TNFRSF8* were expressed at particularly high

**Fig. 4 | IRF4-C99R upregulated genes encompass cHL hallmark genes in a non-canonical AICE2-dependent manner. a** Comparison of fold changes between IRF4-C99R-induced genes in mouse splenic B cells with differentially expressed genes of Hodgkin and non-Hodgkin cell lines based on microarray gene expression analyses. The Spearman correlation between the log2 fold changes of cHL (L428, L1236, KM-H2, HDLM-2)/non-cHL (REH, NAMALWA, SU-DHL-4) cell lines versus C99R/MIG was −0.25, with a P value less than $10^{-5}$. The number of genes compared was 434, comprising the most differentially expressed genes from microarray analyses between cHL and non-cHL cell lines. Note, that IRF4-C99R-induced genes include the known HL-hallmark genes *GATA3*, *CCL5* (*RANTES*), and *TNFRSF8* (*CD30*) (red rectangle). **b** Left, UCSC Genome Browser screenshot of REH, GM12878, KM-H2$^{IRF4-WT}$ and L428$^{IRF4-C99R}$ DHSs (red) as well as IRF4 (blue) and JUNB (green) ChIP peaks at the *GATA3* gene locus. L428$^{IRF4-C99R}$-specific ChIP peaks used for EMSA analyses are indicated by gray bars and designated as *GATA3*Peak_1, *GATA3*Peak_2, and *GATA3*Peak_3. Right, HEK293 cells were control transfected (−), or transfected with IRF4-WT, IRF4-C99R, JUNB and BATF, or combinations thereof, as indicated. Nuclear extracts were analyzed for DNA-binding activity at WT and ISRE-mutated *GATA3*Peak_1, *GATA3*Peak_2 and *GATA3*Peak_3 sites. EMSA data show one out of three independent experiments. **c** Information content (in bits; *y* axis) of half-ISRE motifs within motifs identified in the ChIP-seq data of L428$^{IRF4-C99R}$ (purple) or KM-H2$^{IRF4-WT}$ (green) cells present in any motif (i.e., all; left, WT *n* = 8; C99R *n* = 10), or annotated as AICE BP1–3 (center, WT *n* = 4; C99R *n* = 6) or ISRE (right, WT *n* = 4, C99R *n* = 4). The logos underneath of each plot represent the summary of the half-ISRE motif for each condition. All ChIP-Seq values are from two averaged replicates per condition. All box-whisker blots represent the median (central line), 25th–75th percentile (bounds of the box) and minimum–maximum (whiskers). Statistical significance was computed using the Welch's *t* test (one-tailed). **d** HEK293 cells were transfected with reporter construct encompassing *GATA3*Peak_1 together with AP-1 (JUNB and BATF) and IRF4 variants. Luciferase activity is shown as fold activation compared to that of control transfected cells (far left), which is set as 1. Statistics were derived from *n* = 3 independent transfections, the mean ± SEM is shown. *P* values were determined by two-tailed unpaired Student's *t* test. One out of three independent experiments is shown. **e** Comparison of gene expression changes between cell lines harboring C99R mutation (C99R/WT cHL) versus fold change between HL and non-Hodgkin cell lines based on RNA-seq analyses. The Spearman correlation between the log2 fold changes of cHL/non-cHL versus C99R cHL/WT cHL was 0.88, with a P value of less than $10^{-5}$. C99R-containing cHL cell lines were L428 and U-HO1, while the WT cell lines were L1236, KM-H2, HDLM-2 and L540Cy. Non-cHL cell lines were REH, NAMALWA, SU-DHL−4 and BJAB. Source data for (**b**–**d**) are provided in the Source Data file.

levels in the cell lines with IRF4-C99R (Fig. 4e). Thus, C99R is the primary inducer of the Hodgkin expression program in C99R bearing cells.

## Discussion

The work described here presents evidence for a somatic mutation-induced fundamental shift in TF DNA-binding specificity and motif recognition, caused by a Cys-to-Arg substitution in the α3-recognition helix of the DNA-binding domain of the IRF4 protein (IRF4-C99R). This mutation is a hallmark of human lymphoma exhibiting perturbed B-cell identity, cHL and PMBCL. IRF4-C99R is to a large extent unable to regulate canonical IRF4 target genes, including those coordinating terminal B-cell differentiation[40], and profoundly blocks plasmablast formation, i.e., terminal B-cell differentiation. Instead, it enforces an altered, disease-specific gene expression program driven by preferred binding to canonical AICE2 sites, and by compelling neomorphic binding to non-canonical AICEs. The IRF4-C99R-mediated altered DNA-binding preferences were not restricted to its interaction with JUNB/BATF-AP-1 heterodimers, but were also observed with other JUN and BATF family members, specifically c-JUN and BATF3, which are also known to be deregulated in cHL[45,54,55]. How direct protein-protein interactions, which have been suggested for binding to AICEs[16,56], contribute to the cooperative binding patterns described here remains to be investigated in future studies.

Functionally, IRF4-C99R combines distinct LOF- and gain-of-function (GOF) properties, which are directly related to its switch in DNA-binding specificities. The block in B-cell differentiation in cHL involves genetic and epigenetic alterations as well as lineage-inappropriate gene expression[30,32,33,57,58]. Our data demonstrate how IRF4-C99R's LOF properties contribute to the block in differentiation, one of the hallmarks of malignant transformation[59]. In line, IRF4-C99R is mostly associated with cHL and PMBCL and is rarely found in other mature B-cell malignancies with maintained B-cell phenotype. Whether IRF4-C99R actively represses lineage differentiation remains to be clarified. On the other hand, IRF4-C99R's GOF properties are exemplified by the distinct activation of lymphoma-specific gene expression, including *GATA3*, *CCL5*, and *TNFRSF8*. Expression of these genes is stongly associated with the cHL phenotype and they play important roles not only for the tumor cells themselves but also for their interaction with the microenvironment[31,53].

Overall, our data demonstrate that AICE motifs are not only key regulatory elements of cellular differentiation and activation processes in immune cells such as T cells or dendritic cells[26–28,56], but that they can non-canonically interact with mutant TFs to establish malignancy-associated gene expression. The arginine mutation-induced shift in DNA-binding and gene regulation highlights the critical role of arginine residues in determining interactions with the DNA interface[11,12]. Moreover, we provide a prominent example how nucleotide sequences flanking the TF core DNA-binding motif modify TF:DNA interactions. Such an Arg-induced shift in TF binding specificity to distinct CEs is difficult to predict with current methodologies and distinguishes IRF4-C99R from most other mutations affecting TF:DNA interactions reported previously[4–6,60,61] and might also operate in other diseases. Indeed, within the framework of the IRF4 International Consortium, we recently reported a complementary recurrent heterozygous p.T95R mutation targeting IRF4's DNA-binding domain as the cause of an autosomal dominant combined immunodeficiency (CID)[62]. However, while both the mutated C99R- and T95R-IRF4 proteins have lost the ability to regulate canonical IRF4 target genes and to exert IRF4's physiological function to coordinate plasma cell differentiation, they also remarkably differ. IRF4-C99R is a somatic mutation, IRF4-T95R a germline mutation. In addition, whilst IRF4-T95R shows an overall broadly increased DNA-binding affinity to canonical and non-canonical DNA motifs, IRF4-C99R displays a unique and distinct loss-and-gain pattern of DNA-binding and regulates different gene sets. We suggest to name such diseases 'Mutation-Induced Neomorphic Transcription factor Binding' (MINTraB)-induced diseases.

Disease-causing TF activities can in principle be therapeutically targeted, which has been shown for a few examples like MYC or NOTCH1 (refs. 63–65). For this purpose, small compound or peptide-based inhibitors have been reported. Thus, the data presented here open the possibility of designing inhibitors that specifically block the neomorphic DNA-binding activity of a mutant TF, without modulating the activities of its normal counterpart.

## Methods

### Statement on ethical regulations

We confirm that our study complies with all relevant ethical regulations. Human patient material was analyzed retrospectively. Samples were provided by the University Cancer Center Frankfurt (UCT, Germany), the Hematopathology Section of Christian-Albrechts-University Kiel (Germany), and the Lymphoma Reference Centre at the Institute of Pathology, University of Würzburg (Germany). We used archived anonymized specimens from patients with diagnosed cHL. The use of human material was approved by the institutional review boards and local Ethics Committees of Charité and University Cancer Center Frankfurt (SHN-06-2018; 15-6184-BO; EA2/087/16), and an individual informed consent for the use of these anonymized specimens is not required. All animal experiments were approved by the

local authority Landesamt für Gesundheit und Soziales (LAGeSo; X9027/11).

## Sex and gender reporting

Classic Hodgkin lymphoma, the main lymphoma entity for which we show data in our manuscript, shows only minor differences in its sex distribution (ratio male:female–1,3: 1), with an even equal distribution between men and women in young adulthood[66,67]. Thus, a clear-cut sex-based phenotypic bias in unlikely. In our study, we thus did not specifically select for sex or perform gender-based analyses. Primary lymphoma samples were selected on a random base for the immunohistochemistry analyses. Given the large number of cases analyzed, it is conceivable that our data reflect the overall distribution of male and female cases within the general population. Selection of materials for lymphoma single-cell analyses as well as of the cell lines was primarily determined by the limited availability of respective materials within the community.

## Cell lines, culture conditions, and transfections

HRS [L428, L1236, KM-H2, L591 (EBV⁺), U-HO1 (all of B-cell origin); HDLM-2, L540, L540cy (all of T-cell origin)], pro-B lymphoblastic leukemia (REH), Burkitt's lymphoma (NAMALWA, BL-60, BJAB), diffuse large B-cell lymphoma (SU-DHL−4), and HEK293 cell lines were cultured as previously described[32,68]. Cell lines used in our study were obtained from the German Collection of Microorganisms and Cell Cultures (DSMZ), the American Tye Culture Collection, and other investigators (L1236 and L591 from V. Diehl, Cologne, Germany; L540cy from A. Engert, Cologne, Germany; BJAB from P. Krammer, Heidelberg, Germany). Cell lines were regularly tested negative for mycoplasma contamination, and their authenticity was verified by STR fingerprinting. For preparation of nuclear extracts for DNA-binding studies, HEK293 cells were transfected by electroporation in OPTI-MEM I using Gene-Pulser II (Bio-Rad) with 960 µF and 0.18 kV with 10 µg pcDNA3-FLAG-JUNB, 10 µg pcDNA3-FLAG-BATF, or increasing amounts, ranging from 0.5 to 10 µg, of the respective pHEBO-IRF4 variants. For analysis of luciferase activity, HEK293 cells were transfected with 15 µg of pGL3_GATA3-3P_AICE_long reporter construct, together with 150 ng pRL-TKLuc as an internal control, where indicated together with 5 µg pcDNA3-FLAG-JUNB, 5 µg pcDNA3-FLAG-BATF, or 40 µg of the respective pHEBO-IRF4 variants. Forty-eight hours after transfection, the ratio of the two luciferases was determined (Dual luciferase kit, Promega). For generation of inducible BJAB cells, cells were electroporated with 40 µg of pRTS1-IRF4-WT or -IRF4-C99R or pRTS1 control plasmid in OPTI-MEM I using Gene-Pulser II with 50 µF and 0.5 kV. Twenty-four hours after transfection, 28 µg/mL Hygromycin B (Sigma-Aldrich, Taufkirchen, Germany) were added. After 21–28 days of culture in the presence of Hygromycin B, cells were suitable for functional assays. The respective IRF4 variants were induced by the addition of 100 ng/mL doxycycline (D9891; Sigma-Aldrich).

## Preparation of whole cell and nuclear extracts, immunoblotting, and electrophoretic mobility shift assays (EMSA)

Preparation of whole cell and nuclear extracts as well as immunoblotting and EMSA were performed as previously described[32,33,68]. For EMSA analyses, we used 3–5 µg nuclear extracts per lane. EMSA buffer contained 10 mM HEPES, pH 7.9, 70 mM KCl, 5 mM dithiothreitol, 1 mM EDTA, 2.5 mM MgCl₂, 4% Ficoll, 0.5 mg/ml BSA, 0.1 µg/ml poly-deoxyinosinic-deoxycytidylic acid (poly[(dI)•(dC)]). The double-stranded oligonucleotides used for EMSA are indicated in Supplementary Data Table 4. After annealing, oligonucleotides were end-labeled with [α-³²P]dCTP with Klenow fragment. Positions of the complexes were visualized by autoradiography. Antibodies used for supershift analyses and for immunoblotting are indicated in

Supplementary Table 5. If not validated by the manufacturer, we validated antibodies with respective positive and negative controls (cell lines, transfected cells).

## DNA constructs

The pHEBO-IRF4-HAtag expression construct and its control pHEBO-CMV-HAtag were kindly provided by L. Pasqualucci (New York). The R98A, C99A, C99R, and S104T mutations were introduced by use of the QuikChange Multi Site-Directed Mutagenesis Kit (Stratagene) into the pHEBO-IRF4-HAtag expression construct according to the manufacturer's recommendations and by use of primers indicated in Supplementary Table 4. For the retroviral transduction experiments of C57BL/6 splenic B cells, the coding sequences for human *IRF4* (WT, C99R, R98AC99A) were amplified from the pHEBO-constructs using the IRF4_XhoI_forw 5′-ACCTCGAGGCCACCATGAACCTGGAGGGCG GCGGCCGA-3′ and IRF4_EcoRI_rev 5′-ACGAATTCTTAAGGCCCTGG ACCCAAAGAAGCGTAATC-3′ primers and cloned in front of the IRES sequence of the MSCV-IRES-GFP (MIG) plasmid (kindly provided by F. Rosenbauer, Münster, Germany) via XhoI and EcoRI. For the pRTS1-based inducible expression constructs[69] of the IRF4-WT and IRF4-C99R variants, *IRF4-WT* and *IRF4-C99R* were amplified using the respective pHEBO-*IRF4* expression constructs as templates. The amplified *IRF4-WT*- and *IRF4-C99R*-products were ligated via XbaI into pUC19-Sfi, respectively, and mobilized by SfiI digestion for cloning into pRTS1. For the pcDNA3-FLAG-JUNB expression construct, full-length human *JUNB* was amplified from cDNA of the human L428 cell line, and cloned via BamHI and XhoI into pcDNA3-FLAG (Invitrogen). Full-length human *c-JUN* (*JUN*) was amplified from cDNA of the human cell line L1236 by use of primers JUN_FLAG_BamHI s 5′-GCGGATCCACTGCAAAGAT GGAAACG-3′ and JUN_STOP_XhoI as 5′-GCCTCGAGTCAAAATGTTTGC AACTG-3′, and cloned via BamHI and XhoI into pcDNA3-FLAG. pcDNA3-based expression constructs for BATF and BATF3 were previously described[54]. For cloning of the pGL3_*GATA3*-3P_AICE_long reporter construct encompassing *GATA3*Peak_1, DNA from My-La cells was amplified by use of primers GATA3_AICE_KpnI s 5′-GCGGTACCATA CAGACCCTTCCAGCCAC-3′ and GATA3_AICE_XhoI as 5′-GCCTCGAG AACAGATGTGGGGAGTCAGA-3′ and cloned via KpnI and XhoI into the multiple cloning site (MCS) of pGL3 (Promega). All constructs were verified by sequencing.

## Sanger sequencing (cell lines)

Primer sequences for the validation of *IRF4* mutations IRF4-C99R and S104T identified by whole exome sequencing in cHL cell lines were designated using the Primer3 software (version 4.1.0; http://frodo.wi. mit.edu/primer3/) (Supplementary Table 4). cDNA for RT-PCR was synthesized using the Maxima First Strand cDNA Synthesis Kit (Thermo Scientific). Sanger sequencing was performed according to standard procedures.

## Laser microdissection and PCR analyses of primary HRS cells

Tissue samples used for laser microdissection were provided by the University Cancer Center Frankfurt (UCT; Germany) and by the Hematopathology Section of Christian-Albrechts-University Kiel (Germany). Written informed consent was obtained from all patients in accordance with the Declaration of Helsinki, and the study was approved by the institutional review board and local Ethics Committee of University Cancer Center Frankfurt (SHN-06-2018; 15-6184-BO). Pools of 10 HRS cells or pools of 10 non-tumor cells, and membrane sections without tissue as controls were laser-microdissected as previously described[70]. Following digestion with proteinase K for 3 h at 55 °C and heat inactivation for 10 min at 95 °C, a semi-nested, two-rounded PCR with exon-spanning primers was performed to amplify exon 3 of IRF4. PCR products were separated on a 1% agarose gel. Gel-purified products were sequenced on an ABI3130 (Applied Biosystems) and evaluated with SeqScape software v2.5 (Applied Biosystems). For

the assessment of mutations, forward and reverse sequences were mandatory. Primer sequences were (always 5′–3′): IRF4_E3_fw 5′-TCG TGCCACTGTACTCTAGCC-3′; IRF4_E3_rv1 5′-ATCTGGCTGCCTCTGTT AGGT-3′; IRF4_E3_rv2 5′-AGCTAGAAAGTGATGCTCAGAATG-3′; IRF4_E3_fw_II 5′-AGTTCCGAGAAGGCATCGAC-3′; IRF4_E3_rv1_II 5′-AT TGGCTCCCTCAGGAACAA-3′; IRF4_E3_rv2_II 5′-TGTACGGGTCTGAG ATGTCCA-3′. For DNA from frozen tissue sections, the primers IRF4_E3_fw and IRF4_E3_rv2 were used in the first round of PCR (product size 389 bp), and the primers IRF4_E3_fw and IRF4_E3_rv1 in the second round (product size 346 bp). For DNA from paraffin sections, primers IRF4_E3_fw_II and IRF4_E3_rv1_II were used in the first round (fragment size 160 bp), and primers IRF4_E3_fw_II and IRF4_E3_rv2_II in the second round (fragment size 129 bp). PCR conditions were 98 °C 4 min, 40 cycles of 98 °C 30 s, 62 °C 20 s, 72 °C 20 s, final elongation 72 °C 3 min.

### IRF4 mutation analysis in PMBCL patients

To generate a custom cRNA bait library (SureSelect, Agilent Technologies) for targeted gene capture, a total of 106 genes (including IRF4) that have been reported to be affected by genetic aberrations in PMBCL were selected. To ensure high quality, only samples that had a coverage of 100× in ≥80% of the exonic regions were included. The median and mean sequencing coverages were 830× and 666×, respectively. Variant calling and filtering was performed as described earlier[71] with the following adaptations as no germline controls were included: (i) 10%_posterior_quantile >0.1; (ii) 10%_posterior_quantile(realignment) >0.1; (iii) VAF for synonymous and nonsynonymous SNVs <0.45, >0.55, and >0.95 for regions that were not affected by SCNA. Over 3000 mutations were extensively inspected for artifacts and mapping errors through visual inspection with the Integrative Genomics Viewer (IGV). A detailed description of the PMBCL patient cohort, applied sequencing workflow, and corresponding bioinformatical analysis are described in ref. 72, and in Noerenberg et al. (J Clin Oncol, in press). This study was conducted in accordance with the Declaration of Helsinki. The protocol was approved by the local ethics review committee of the Charité−Universitätsmedizin Berlin (EA2/087/16) and of every participating center.

### IRF4 shRNA-mediated cytotoxicity assay of L428 cells

For efficient retroviral transductions, L428 cells were engineered to express a murine ecotropic receptor as previously described[73]. In addition, the cells were also engineered to express a bacterial tetracycline repressor allowing doxycycline-inducible small-hairpin RNA (shRNA) or cDNA expression. The retroviral transduction experiments, shRNA-mediated RNA interference and cytotoxicity assays were performed as described elsewhere[22,73–75]. In brief, to assess toxicity of an shRNA, retroviruses that co-express green fluorescent protein (GFP) were used as described[22,73–75]. Flow cytometry was performed two days after shRNA transduction to determine the initial GFP-positive proportion of live cells for each shRNA. Subsequently, cells were cultured with doxycyline (40 ng/ml) to induce shRNA expression, and the proportion of GFP-positive cell was measured at the indicated time points. The GFP-positive proportion at each time point was normalized to that of the negative control shRNA and further normalized to the day two fraction. The targeting sequence of IRF4 shRNAs #1 and #2 were 5′-CCGCCATTCCTCTATTCAAGA and 5′-GTGCCATTTCTCAGGGAAGTA as described[20,22]. As a negative control shRNA, a previously described shRNA against MSMO1 was used[22]. Each shRNA experiment was reproduced at least two times. For the IRF4 rescue experiments, IRF4 (NM_002460.3) single mutant IRF4$^{C99R}$ and double mutant IRF4$^{C99RS104T}$ cDNAs were created and the experiment was performed as previously described[20,22]. In brief, to assess rescue effect of an IRF4 cDNA, L428 cells were transduced with an IRF4#1 or #2 shRNA, followed by retroviral ectopic expression of either an empty vector or an IRF4 cDNA that co-expresses GFP. We compared cell growth for each overexpression relative to the growth for the empty vector which is normalized to the

100% line, and further normalized to the day two fraction. Each experiment was reproduced at least two times. Combining the four curves (both shRNAs and their replicates) for each cDNA, aggregated curves show mean viabilities (markers) ± standard errors (transparent tunnels). At day 11, we statistically compared with 100%, i.e., with our null hypothesis for zero rescue effect (one-sample one-tailed t tests).

### Immunohistochemistry

Formalin-fixed, paraffin-embedded tissue specimens from cases diagnosed as classic Hodgkin lymphoma (30 cHL mixed cellularity subtype; 30 cHL nodular sclerosis subtype; 30 cases lymphocyte-rich subtype) were retrieved from the files of the Lymphoma Reference Centre at the Institute of Pathology, University of Würzburg, Germany. For this retrospective study we used archived anonymized tissue specimens, and there was no participant compensation. From each paraffin block 2-μm sections were cut and subjected to immunohistochemical stainings. Immunostains were performed in an automatic immunostainer using program ER2 (Bond III, Leica Biosystems, Nussloch, Germany) using the manufacturer's protocols and detection reagents. Detection of IRF4 employed the monoclonal antibody MUM1P (M725929; dilution 1:400; DAKO/Agilent, Waldbronn, Germany).

### Cloning and purification of recombinant proteins

Codon-optimized sequences encoding the DNA-binding domains of human BATF (AA 28–87) and JUNB (AA 269–329) were cloned into pMAL-C2X (NEB). The sequence encoding human IRF4 DBD (AA 20–139) was cloned into pGEX6P1 (Cytivia). IRF4 mutations were introduced by QuikChange Site-Directed Mutagenesis Kit (Agilent) according to the manufacturer's recommendations. All constructs were verified by sequencing. Plasmids were separately transfected into BL21-DE3-Rosetta (Novagen). Proteins were expressed overnight at 18 °C in TB medium (Melford) after induction with 40 mM Isopropyl-β-D-thiogalactopyranosid (IPTG). Cells were resuspended in 50 mM HEPES pH 7.5, 300 mM NaCl, 2.5 dithiothreitol (DTT), 1 μM DNase, 200 μM Pefablock (Carl Roth) and lysed in a microfluidizer (Microfluidics). Eluates containing MBP-fusions were applied to 5 mL amylose resin (NEB) columns and extensively washed with 20 mM HEPES pH 7.5, 150 mM NaCl, 2.5 mM DTT. Proteins were eluted in the same buffer containing additional 10 mM maltose. Eluates containing GST-IRF4 protein were applied to a 5 mM GSH sepharose (Cytivia) column and extensively washed with 20 mM HEPES pH 7.5, 150 mM NaCl, 2.5 mM DTT. Proteins were eluted in the same buffer conaining additionally 20 mM glutathione (pH 7.5) (Sigma-Aldrich). GST was removed by the addition of PreScission Protease in a ratio of 1:100. All proteins were separately concentrated using 10 kD cut-off Amicon Ultra-15 Centrifugal filters (Millipore) and applied to a final gel filtration run on a Superdex 75 column (Cytivia) using 20 mM HEPES, pH 7.5, 150 mM NaCl, 2 mM DTT as running buffer. Peak fractions containing the protein of interest were concentrated and flash-frozen in small aliquots.

### DNase-seq

DNaseI-seq was essentially performed as previously described[68] with slight modifications. Briefly, cells were washed and resuspended at $10^8$ cells/ml in ice-cold ψ buffer (11 mM KPO$_4$, pH 7.4, 108 mM KCl, 22 mM NaCl, 5 mM MgCl$_2$, 1 mM CaCl$_2$, 1 mM dithiothreitol, 1 mM ATP). 1 Mio REH, KM-H2 or L428 cells were treated with 12 U/mL DNaseI (Worthington) for 3 min at 22 °C. Digestion was stopped with the addition of 200 μl lysis buffer (100 mM Tris, pH 8.0, 40 mM EDTA, 2% SDS, 200 μg/ml proteinase K) overnight at 37 °C. DNase digestion efficiency was checked via low-voltage overnight electrophoresis (10 V) on a 0.5% TAE agarose gel. Short-fragment size selection was performed by cutting out gel bands between 100–200 bp and subsequent purification using the QiaQuick gel extraction kit (Qiagen) according to the manufacturer's instructions. Library preparation was performed using

the KAPA hyperprep kit (Roche) following the manufacturer's guidelines. Library quality was checked via qPCR using TBP, ACTB and gene desert control oligonucleotides[76]. Libraries were sequenced at 400 million reads per library in single-end mode on separate lanes using an Illumina HiSeq 2000 system according to the manufacturer's instructions.

## ChIP-seq

ChIP was performed as previously described[77] using double-crosslinking. Cells were resuspended at $3.3 \times 10^6$ cells/mL in PBS and first crosslinked with 8.3 µl/ml DSG (Sigma) for 45 min at room temperature, subsequently washed 4× and crosslinked in 1% formaldehyde for 10 min at room temperature, with both crosslinking methods entailing sustained tube rotation. Crosslinking was quenched in 0.2 M glycine and cells were washed 2×. Cells were lysed in *Buffer A* (10 mM Hepes, 10 mM EDTA, 0.5 mM EGTA, 0.25% Triton X100), then in *Buffer B* (10 mM Hepes, 200 mM NaCl, 1 mM EDTA, 0.5 mM EGTA, 0.01% Triton X100), at $1 \times 10^7$ cells/ml and 4 °C with rotation for 10 min for both stages. Nuclei were resuspended at $2 \times 10^7$ cells/ml in 4 °C *IP Buffer I* (25 mM Tris, 150 mM NaCl, 2 mM EDTA, 1% Triton X100, 0.25% SDS) and sonicated in $6 \times 300$ µl per reaction using a Picoruptor sonicator (Diagenode) at 240 W with 30 cycles of 30 s on, 30 s off at 4 °C. Cell debris was pelleted via 10 min $16{,}000 \times g$ centrifugation and diluted in *IP Buffer II* (8.33 mM Tris, 50 mM NaCl, 6.33 mM EDTA, 0.33% Triton X100, 0.0833% SDS, 5% glycerol final concentration). 5% of chromatin was saved as input control. Immunoprecipitation was carried out overnight using Maximum Recovery tubes (Axigen) with rotation at 4 °C in 50 µl PBS + 0.02% Tween 20 with 15 µl protein G dynabeads that were washed, blocked with 0.5% BSA and conjugated with either IRF4 (sc-6059-X, Santa Cruz) or JUNB (sc–46-X, Santa Cruz) antibodies for 4 h ar 4 °C with rotation. Beads were subsequently washed on ice by magnetic separation using 1× PBS + 0.02% Tween 20, 2× *Wash Buffer 1* (20 mM Tris, 150 mM NaCl, 2 mM EDTA, 1% Triton X100, 0.1% SDS), 1× *Wash Buffer 2* (20 mM Tris, 500 mM NaCl, 2 mM EDTA, 1% Triton X100, 0.1% SDS), 1× with *LiCL Buffer* (10 mM Tris, 250 mM LiCl, 1 mM EDTA, 0.5% NP40, 0.5% Na-deoxycholate), 2× with *TE/NaCL Buffer* (10 mM Tris, 50 mM NaCl, 1 mM EDTA). Beads were eluted using $2 \times 50$ µL *Elution Buffer* (100 mM NaHCO₃, 1% SDS) with shaking for 15 min at RT, and eluates were pooled. Chromatin was reverse-crosslinked overnight at 65 °C in *Elution Buffer* + 200 mM NaCl, followed by 100 µg/ml RNase A and 0.25 mg/ml proteinase K digestion for 1 h at 37 °C and 55 °C, respectively. DNA was purified via phenol chloroform extaction. ChIP efficiency was checked using IL3–40, CSF1R FIRE, and gene desert control oligonucleotides[76]. Library preparation was performed using the KAPA hyperprep kit (Roche) following the manufacturer's guidelines. Libraries were sequenced in single-end mode at 50 million reads per library using an Illumina HiSeq 2000 system according to the manufacturer's instructions.

## Single-molecule fluorescence microscopy

(A) Cloning of IRF4 plasmids for fusion proteins: cDNAs encoding human IRF4 and IRF4-C99R were cloned into the LV-tetO-HaloTag plasmid using EcoRI and XbaI restriction sites and One Shot Stbl3 chemically competent E. coli (Thermo Fisher Scientific, USA)[78]. Coding regions of the plasmids were verified by Sanger sequencing. (B) Generation of stable cell lines: Lentiviral transduction was used to generate HeLa cells which stably express IRF4-WT- or IRF4-C99R-HaloTag fusion proteins[78]. In brief, HEK293T cells were transiently transfected with psPAX2 (Addgene #12260), PMD2.G (Addgene #12259), and the respective pLV-tetO IRF4-HaloTag variants using JetPrime (PolyPlus). Supernatants containing viruses were harvested through a 0.45 µm filter after 48 h. HeLa cells were infected at 37 °C and 5% CO2 for 72 h. (C) Preparation of cells for imaging: One day before imaging, cells were seeded on a heatable glass bottom dish (DelaT, Bioptechs), and 15 min prior to imaging 3 pM silicon rhodamine (SiR) HaloTag ligand (kindly

provided by K. Johnson, Heidelberg, Germany) were added according to the HaloTag staining protocol (Promega). Thereafter, cells were washed with PBS and placed for 30 min at 37 °C and 5% CO₂ in DMEM. Before imaging, cells were washed three times with PBS and imaged in 2 mL OptiMEM. (D) Microscope setup: A custom-built fluorescence microscope for single-molecule fluorescence imaging was used as described[48]. (E) Interlaced time-lapse illumination and data analysis: Cells were illuminated with a highly inclined light beam[79] using an interlaced time-lapse illumination scheme[48]. In ITM, we repeated a pattern of two consecutive images with 50 ms camera integration time followed by a dark time of 2 s. Localization of fluorescent molecules within an image and tracking of molecules across consecutive images was performed by use of TrackIt v1.0.1 (ref. [80]). Detection and tracking parameters were: threshold factor' 3, 'tracking radius' 2, 'min. track length' 2, 'gap frames' 0, 'min. track length before gap frame' 0. Molecules only detected within a single image were classified as unbound, the ones detected in two consecutive images within an area of 0.35 µm² as short-bound, and those tracked over at least one dark time as long-bound. For each imaged cell, the ratio of all bound molecules (including short- and long-bound molecules) to all molecules (including long-, short-, and unbound molecules) and of long-bound molecules to all molecules was calculated. The significance between IRF4-WT and IRF4-C99R was tested with an unpaired, non-parametric *t* test (Mann–Withney test) using Graphpad prism 9.0.1.

## Reference-free DNA modeling and IRF4 docking studies

The structural modeling is designed to provide insight as to how the IRF4-C99R mutation can influence binding to the different DNA motifs and to complement the functional data observed in this study. To model the structural basis for the interaction of IRF-WT or IRF4-C99R with different DNA elements, unbiased random docking and interaction studies were examined using HADDOCK 2.2 (ref. [81]). The initial structures of IRF4-WT and the ISRE DNA were obtained from our previous crystal structure (PDB: 7JM4). In generating the annealed AICE1 DNA motif, template-based free annealing and ternary structure of the DNA fragment were obtained using HNADOCK DNA program[82]. As the option to generate a DNA structure does not exist in HNADOCK, ssDNA structures were initially generated in PyMOL v2.5 and then imported, and energy was minimized in HNADOCK to generate the dsDNA. Deprived annealing of DNA ends was noted due to low Tm in the modeling at the 5' or 3' ends. All docking studies and modeling were performed using standard HADDOCK 2.2. The HADDOCK program differs from ab initio docking methods by utilizing information from known or predicted protein interfaces for ambigous interaction restraints (AIRs) and utilizes flexible docking. The default HADDOCK program was used without additional restraints. No MD simulations were used, and flexible docking was performed as a default option in HADDOCK with default energy minimization. Furthermore, all the hydrogen atoms in the initial PDB structures were retained, thereby simulating more realistic results. Using the Kyte-Doolittle mode in the HADDOCK 2.2, in silico water molecules were added while docking the IRF4 with DNA to distinguish any water-mediated solvation contacts. As shown in the modeling analysis report in Supplementary Table 3, to assess the quality and confidence in the modeled structures, several modeling outcome parameters were analyzed as per the HADDOCK guidelines (https://www.bonvinlab.org/software/haddock2.2/analysis/). Primarily the lower values of HADDOCK score, Z-score, Van der Waals and electrostatic energy were all considered to denote better models. In fact, for all parameters, apart for the buried interface and desolvation energy, lower values signified greater confidence in the proposed model. The modeled structures and the interaction interface of IRF4-WT or IRF4-C99R with different DNA elements (ISRE or AICE1) were assessed for their quality using HADDOCK. Analysis of output parameters of 5 different clusters obtained from the docked models (HADDOCK 2.2) was used to validate that the models were true and

consistent across all clusters. The binding free energies were also taken into consideration in selecting the best possible models. Further validation and refinement was undertaken by ensuring that the residues occupied Ramachandran favored positions using Coot (https://www2.mrc-lmb.cam.ac.uk/personal/pemsley/coot/). The figures were generated using PyMol v2.5.

### RNA-seq of human lymphoma cell lines

RNA-seq analyses of L428, L1236, KM-H2, U-HO1, L591, HDLM-2, L540, L540cy, REH, NAMALWA, BJAB and SU-DHL-4 cells was performed in duplicates. In brief, barcoded mRNA-seq cDNA libraries were prepared from 600 ng of total RNA using Illumina's TruSeq Stranded RNA Sample Preparation Kit. mRNA was isolated using Oligo(dT) magnetic beads. Isolated mRNA was fragmented using divalent cations and heat. Fragmented mRNA was converted into cDNA using random primers and SuperScriptII (Invitrogen). This was followed by second-strand synthesis. cDNA was repaired and 3′ adenylated. 3′ single T-overhang Illumina multiplex specific adapters were ligated on the cDNA fragments, and these fragments were enriched by PCR. All cleanups were done using Agencourt XP magnetic beads. Barcoded RNA-seq libraries were clustered on the cBot using the Truseq PE cluster kit V3 using 10 pM and 2 × 50 bps were sequenced on the Illumina HiSeq2500 using a Truseq SBS V3 kit.

### Generation of retroviral particles, mouse B-cell isolation, retroviral transduction

By use of calcium-phosphate buffer, 10 μg of retroviral plasmids (MSCV-based) encoding human IRF4-WT, IRF4-C99R, or IRF4-R98AC99A were transfected into the Plat-E packing cell line[83], together with packaging plasmids pGagpol (10 μg) and pEnv (2 μg) (both courtesy of A. Leutz, Berlin) and 25 μM chloroquin (#6628, Sigma). Thereafter, cells were incubated for 6–8 h at 37 °C and 5% $CO_2$, followed by change of medium to B-cell medium (DMEM high glucose (4,5 g/l) supplemented with 10% FCS, 1% sodium pyruvate, 1% penicillin-streptomycin, 1% HEPES, 1% l-glutamine, 1% non-essential amino acids and 0.05% β-mercaptoethanol) and further cultivation. 48 h after transfection, cell culture supernatants were harvested, filtered (0.45 μm) and frozen at −80 °C. Splenic B cells were isolated from 8- to 12- week-old C75BL/6 mice (originally obtained from Jackson Laboratories) by CD43 depletion with magnetic anti-mouse CD43 microbeads (#130-049-801, Milenyi Biotech) according to the manufacturer's instructions. Purified B cells (density $1 \times 10^6$ cells/ml; $4 \times 10^6$ cells per well) were cultured in the presence of recombinant mouse IL-4 (25 ng/ml; #404-ML, R&D) and LPS (20 μg/ml; #L2880, Sigma) overnight to induce B-cell activation and terminal differentiation. 24 h after isolation, B cells were collected ($300 \times g$, 5 min, 4 °C) and resuspended in B-cell medium supplemented with 8 μg/ml polybrene (#TR-1003, EMD Millipore) at a density of $2 \times 10^6$ cells/ml. To introduce the IRF4 variants, $4 \times 10^6$ B cells per well were plated in 2 ml on 6-well plates that had been coated with RetroNectin (25 μg/ml, 4 °C, overnight; #T100B, Takara), blocked with 2% BSA in PBS (1 h) and pre-loaded with the respective retroviral particles (1 h, 37 °C). Retroviral transduction was performed by the addition of 2 ml of the respective retroviral supernatant and subsequent centrifugation ($800 \times g$, 90 min, 32 °C). 24 h after transduction, B cells were collected ($300 \times g$, 5 min, 4 °C), resuspended in B-cell medium and cultured (density $1 \times 10^6$ cells/ml; $4 \times 10^6$ cells per well) for another 72 h (FACS for RNA-seq, flow cytometric analysis of plasma cell differentiation) in the presence of recombinant mouse IL-4 and LPS. Animal experiments were approved by the local authorities (Landesamt für Gesundheit und Soziales, LAGeSo; X9027/11).

### Flow cytometry of C57BL/6 splenic B cells

Retrovirally transduced B cells were harvested, blocked with TruStain FcX (α-mouse CD16/32; 10 min, 4 °C; #101320, BioLegend) and stained (20 min, 4 °C) with B220-PerCP/Cyanine5.5 (#103235; BioLegend) and CD138-PE (#142504; BioLegend) in PBS, pH 7.2, supplemented with 3% FCS and 1 mM EDTA. Analysis of the samples was performed on a FACSCantoII instrument (BD BioSciences) or sorted on a FACSAria (BD BioSciences). FlowJo software (BD FlowJo, RRID:SCR_008520; v9.9.6) was used to generate plots.

### Bioinformatics analyses of HL RNA-seq, DNase-seq, and ChIP-seq data

**HL cell line RNA-seq processing.** Reads were aligned in paired-end mode to the hg19 genome using STAR v2.3.0 (ref. [84]) using --outSAMattributes Standard --outSAMunmapped None --outReadsUnmapped Fastx --outFilterMismatchNoverLmax 0.02 as parameters. Counts were obtained using featureCounts v2.0.0 (ref. [85]) with -p -B -C -Q 10 --primary -s 0 as parameters. Normalization and differential gene expression analysis were performed using DESeq2 v1.14.1 (ref. [86]) using the standard analysis protocol, performing variance stabilization transform normalization. Gene set enrichment analyses were performed using GSEA v3.0 (ref. [87]).

**DNase-seq and ChIP-seq processing.** Base-calling was carried out using HiSeq Analysis Software v2.0 (Illumina). Reads were demultiplexed using bcl2fastq v2.16.0 (Illumina). As libraries were sequenced in separate lanes, unassigned reads/unreadable indexes were assigned to their respective lane. Reads were subsequently aligned in single-end mode to the hg19 genome using bowtie2 v2.1.0 (ref. [88]) using --very-sensitive-local as a parameter and sorted by coordinate ordering using samtools sort v1.1 (ref. [89]). Peak calling and depth coverage track generation were carried out using macs2 v2.1.0 (ref. [90]) using callpeak -g hs -q 0.001 -B --SPMR --trackline <trackline> as parameters, with --keep-dup all and --keep-dup auto for DNaseI- and ChIP-Seq assays, respectively, to account for high depth sequencing of DNase-Seq libraries. Peak calling yielded 61983, 65612, and 68370 peaks for Reh, KM-H2, and L428 DNase-Seq datasets, respectively, 33082 and 30022 peaks for KM-H2 and L428 IRF4 ChIP-Seq datasets, respectively, as well as 24914 and 24886 peaks for KM-H2 and L428 JUNB ChIP-Seq datasets, respectively.

**DNase-seq and ChIP-seq processing.** For pairwise comparisons, the union of peak summits was obtained as previously described[91] and masked against blacklisted and simple repeat regions[92] using bedtools intersect v2.19.0 (ref. [93]) with -v as a parameter. Corresponding depth coverages were obtained using Homer annotatePeaks v4.6 (ref. [94]) with -hist 10 -ghist -size 2000 as parameters and subsequently $\log_2$ fold-changed ranked on total signal [−100 bp; +100 bp] around peak summits as previously described[77]. Heatmaps were obtained using Java TreeView v1.1.4 (ref. [95]). Venn diagramme overlaps and specific peak populations were computed using ChIPpeakAnno makeVennDiagram v.1.12.0 (ref. [96]), and bedtools intersect with totalTest = <sum of ChIP-Seq peak numbers> and -u as a parameter, respectively. Spearman correlation clustering heatmaps were obtained via gplots heatmap.2 v2.17.0 using total ChIP-Seq signal [−100 bp; +100 bp] around the union of GM12878, KM-H2 and L428 IRF4 ChIP-Seq peak summits. For motif, DNase-Seq and ChIP-Seq average profile significance testing, signal [−200 bp; +200 bp] from summits were averaged per region, split into three classes and tested for significance using t tests. For GSEA analyses, peaks were annotated to the closest gene using bedtools closest using -t first as a parameter; the top 1000 specific peaks sorted by decreasing signal were used. Footprinted motif co-occurrence clustering was performed as previously described[77,91] with specific peaks and 1000 similarly-sized samplings of control peaks being annotated with 16 motifs using Homer annotatePeaks. Intersection matrices were computed using pyBedtools intersection_matrix[97]. Enrichment z-scores were

computed by subtracting mean co-occurrences between the specific peaks and control peaks, dividing by the standard deviation of control peaks.

**Public dataset processing.** Reads for IRF4 and JUNB GM12878 ENCODE ChIP-Seq datasets[98,99] and GM12878 ENCODE DNase-Seq[100] were retrieved from the Sequence Read Archive (SRA) and processed as ChIP-Seq datasets above. L428 and REH DNase-seq datasets generated in this study were complemented in reads from corresponding previously published, lower sequencing depth datasets[68].

**Motifs discovery, average profiles, and heatmaps.** Motif discovery was performed using Homer findMotifsGenome using default parameters. Motif average profiles and heatmaps were generated using Homer annotatePeaks using -hist 10 -ghist -size 2000 as parameters and plotted using Java TreeView. Statistics were done in R v4.0.3.

**Digital genomic footprinting.** Digital genomic footprinting was performed using pyDNase wellington_footprints v0.2.6 (ref. 101) using -A as a parameter, yielding 60,669, 75,813, and 75,755 footprints for REH, KM-H2 and L428 cells, respectively. Individual motif bed files were obtained from union of REH, KM-H2 and L428 footprints annotated for each motif using Homer annotatePeaks -mbed -size given as parameters and subsequently plotted as motif footprint profiles using pyDNase dnase_average_profiles with -A -n as parameters. AICE2 and AICE2[FLIP] motifs were obtained from ref. 47. Motif footprinting scores were obtained using wellington_score_heatmap using -A as a parameter, with scores at the footprint centre being log2 transformed and used for $t$ test significance testing[102].

### Bioinformatic analysis of mouse splenic B-cell and lymphoma cell line RNA-seq data

**RNA-seq of splenic B cells.** RNA prepared from isolated murine splenic B cells transduced with MIG control virus or IRF4-WT, IRF4-C99R, or IRF4-R98AC99A variants was processed by use of the KAPA mRNA Hyper Prep Kit for Illumina Platforms (KK8580; Roche) and KAPA Single-Indexed Adapter Kit (KK8700; Roche). Libraries were sequenced by use of Illumina HighSeq 4000. RNA-seq data from mouse splenic B cells was processed using PiGx-RNA-seq[103] pipeline. In short, the data was mapped onto the GRCm38/mm10 version of the mouse transcriptome (downloaded from the ENSEMBL database[104]) using SALMON (v0.9.1)[105]. The quantified data was processed using tximport (v1.22.0)[106], and the differential expression analysis was done using DESeq2 v1.34.0 (ref. 86). Genes with less than 5 reads in all biological replicates of one condition were filtered out before the analysis. Two groups of differentially expressed genes were defined—a relaxed set containing genes with an absolute log2 fold change of 0.5, and a stringent set containing games with an absolute log2 fold change of 1. The fold change was deemed significant if the adjusted $P$ value was less than 0.05 (Benjamini–Hochberg corrected).

**RNA-seq of lymphoma cell lines.** RNA-seq data of the Hodgkin and non-Hodgkin cell lines was processed using the PiGx-RNA-seq[103] pipeline. In short, the data were mapped onto the GRCh38/hg38 version of the human transcriptome using SALMON (v0.9.1)[105]. Differential gene expression results were integrated with the previously analyzed microarray data from cHL cell lines[107].

**Data integration and visualization.** All analyses were done in R v4.1 using custom scripts. Quantified genes were imported into R using the tximport package (v1.22.6). Splenic B-cell per-sample heatmap was constructed by calculating the Pearson correlation coefficient DESeq2 normalized expression values. The heatmap was visualized using the ComplexHeatmap package (v2.10.0)[108]. The number of stringently differentially expressed genes in each condition was visualized using

UpSet diagrams (UpSetR v1.4.0)[109]. Human and mouse genes were mapped through the orthologous assignment using the ENSEMBL database. Monocyte, B cell, and plasma cell expression profiles were extracted from the ARCHS4 database[110]. Samples with the following keywords in the Sample_source_name_ch1 field were used in the analysis: "granulocyte-monocyte progenitor (GMP) cells", "Blood-derived monocyte", "Bone marrow, plasma cells, WT"; "Splenic B cells, Wild type"; "WT, B cells"; "LPS activated B cells". If multiple samples corresponded to one condition, their expression values were averaged.

**Microarray analyses of inducible BJAB cells.** For microarray gene expression analyses of BJAB cells following tet-induction of IRF4-WT or IRF4-C99R, mRNA was processed by use of the Illumina Total Prep RNA Amplification Kit (AMIL1791; Invitrogen) and use of Human HAT-12_v4 Bead Chips (Affimetrix). The microarray gene expression data from a total of 24 arrays were analyzed in GenomeStudio software (Illumina, Little Chesterford, UK) with background subtraction from experiments that were performed with Tet-inducible BJAB cells expressing Mock control, IRF4-WT or the IRF4-C99R mutant. The raw data output by GenomeStudio was analyzed using the Lumi R package[111] with quantile normalization. The 10% threshold ($P$ value <= 0.1) was applied to all samples. Genes with at least twofold-change in expression (either up or down) were selected in either IRF4-WT versus Mock and IRF4-C99R vs Mock in different time courses. Principle component analysis and hierarchical clustering were carried out using all expressed genes across all replicate samples to show that replicates are highly correlated, and then hierarchical clustering of differentially expressed genes was carried out only for genes associated with at least a twofold change in one condition in either IRF4-WT versus Mock and IRF4-C99R vs Mock. Hierarchical clustering was used with Euclidean distance and average linkage clustering.

### ExplaiNN models and calculation of information content

**Deep-learning models.** Four different ExplaiNN models[49], each with 100 units, were trained on either IRF4-C99R or IRF4-WT ChIP-/DNase-seq data. The architecture of each unit was as follows: • 1st convolutional layer with 1 filter (26 × 4), batch normalization, exponential activation to improve the representation of the learnt sequence motifs[112] and max pooling (7 × 7); • 1st fully connected layer with 100 nodes, batch normalization, ReLU activation and 30% dropout; and • 2nd fully connected layer with 1 node, batch normalization and ReLU activation. For training the models, ChIP-/DNase-seq peaks were resized to 201 bp by extending their summits 100 bp in each direction using BEDTools slop (version 2.30.0)[93]. Negative sequences were obtained by dinucleotide shuffling each dataset using BiasAway (version 3.3.0)[113]. Sequences were randomly split into training (80%), validation (10%) and test (10%) sets using the "train_test_split" function from scikit-learn (version 0.24.2) (https://jmlr.csail.mit.edu/papers/v12/pedregosa11a.html). Models were trained as described in ExplaiNN. Briefly, using the Adam optimizer (https://arxiv.org/abs/1412.6980) and binary cross entropy as loss function, applying one-hot encoding, setting the learning rate to 0.003 and batch size to 100, and using an early stopping criteria to prevent overfitting. Models were also interpreted following the specifications from ExplaiNN. The filter of each unit was converted into a motif by aligning all subsequences activating that filter's unit by ≥50% of its maximum activation value in correctly predicted sequences. The importance of each motif was calculated as the product of the activation of its unit for each correctly predicted sequence activating that unit by ≥50% of its maximum activation value times the weight of the final layer of that unit.

**Information content.** For each ChIP-seq motif annotated as AICE1, AICE2, or AICE2[FLIP], the information content for the four bp corresponding to the half-ISRE site was calculated using Biopython[114].

Summary motifs of the half-ISRE sites in present in these motifs were obtained by aligning the individual 4-mers corresponding to the half-ISRE sites. Statistical significance was computed using the Welch's *t* test (one-tailed) as implemented in SciPy (version 1.7.1)[115].

## Statistics and reproducibility

Statistical analyses were mainly performed in R v4.0.3, R v4.1 and Prism v9.0.2. The correlation of data was determined by the Pearson correlation coefficient. Other data are presented as mean ± SEM or as box-whisker blots showing median, 25th–75th percentile, and minimum–maximum as stated in the respective figure legends. *P* values were determined by two-tailed unpaired Student's *t* test without adjustment for multiple comparisons if not indicated otherwise in the figure legends. No statistical method was used to predetermine sample size. No data were excluded form the analyses apart from technical failures. The experiments were not randomized. The investigators were not blinded to allocation during experiments and outcome assessment.

## Reporting summary

Further information on research design is available in the Nature Portfolio Reporting Summary linked to this article.

## Data availability

Data presented in this study are available at the Gene Expression Omnibus[116] under superseries accession GSE211445. Deposited datasets are DNaseI-Seq: GSE211441; ChIP-Seq: GSE211443; HL and NHL cell line RNA-Seq: GSE211444; BJAB cells with Tet-inducible control, IRF4-WT, and IRF4-C99R Illumina BeadChip HT-12 V4.0 expression arrays: GSE211913. RNA-Seq data of mouse splenic B cells are deposited in ArrayExpress database under ID E-MTAB-12522. High-throughput sequencing data of the PMBCL cohort is in part publicly available[72], and in part deposited (BioProject PRJNA851197 and EGAS00001006452) (Noerenberg et al., J Clin Oncol, in press) but currently only accessible upon request from F. Damm (frederik.damm@charite.de). Source data are provided with this paper.

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

## Acknowledgements

The authors like to thank Natalia Soloch (Poznan), Sabine Werner (Berlin) and Brigitte Wollert-Wulf (Berlin) for excellent technical assistance, and P. Rahn (Berlin) for cell sorting. Support for infrastructure has been provided by the KinderKrebsInitiative Buchholz/Holm-Seppensen. We thank Wolfram Klapper (Kiel) for providing HL tissue samples, and Patrick Sorn (Mainz) and the Team Medical Genomics (TRON gGmbH, Mainz) for RNA-seq processing and data analysis. R.S. and M.G. received funding from the European Union's Horizon 2020 research and innovation program under grant agreement No 952304. O.F. and W.W.W. were supported by grants from the Canadian Institutes of Health Research (PJT-162120), Natural Sciences and Engineering Research Council of Canada (NSERC) Discovery Grant (RGPIN-2017-06824), and BC Children's Hospital Foundation and Research Institute. R.K. and M.-L.H. were supported by the Wilhelm Sander Foundation (2018.101.1). This study was in part supported by grants from the Brigitte and Dr. Konstanze Wegener-Stiftung (#55), the Deutsche Krebshilfe (#70113148, #70113643) awarded to F.D., funds of the Max Planck Society (K150) to R.G., the Deutsche Forschungsgemeinschaft to M.J. and S.M. (MA 3313/2-1 and JA 1847/2-1). Work in the lab of C.B. and P.N.C. was funded by a Blood Cancer

Research UK program grant (15001), the Kay Kendall Leukemia Fund (KKL725) and a studentship donation from Arthur D. Riggs from City of Hope for B.E.-W.

## Author contributions
V.F. and M.G. contributed equally to this work as co-second authors. N.S., P.C., V.F., M.G., and O.F. designed and performed experiments, interpreted data and wrote the manuscript; N.V., M.G.C., and S.S. performed and interpreted structure modeling; S.A.A. analyzed and interpreted microarray data; M.A.W., M.-L.H., S.H., and R.K. designed, performed, and interpreted HRS single-cell analyses; I.A. and A.R. performed and interpreted IHC analyses; F.D. and D.N. performed and interpreted PMBCL analyses; O.D. and SG designed and performed production of recombinant proteins; J.C.M.G. and A.Re. designed, performed and interpreted single-molecule fluorescence microscopy; T.B. analyzed RNA-seq data; E.K. and S.L. performed experiments and interpreted data; U.P., W.W., and M.C. performed experiments, interpreted data and contributed to writing of the MS; A.F. interpreted the data; B.E.-W., L.H., P.C., and N.O. designed, perfomed and interpreted DNase and ChIP experiments; A.W., W.X., M.Gr., and G.L. designed, performed and interpreted shRNA experiments; K.S., K.R., G.L., A.A., and W.W.W. interpreted the data and contributed to writing of the manuscript; P.N.C., C.S., R.S., R.K., R.G., M.J., and C.B. designed research, interpreted data and wrote the manuscript, S.M. designed research, interpreted data, wrote the manuscript and supervised the project. All authors discussed the results and commented on the manuscript.

## Funding

## Competing interests
The authors declare no competing interests.

## Additional information

[1]Max-Delbrück-Center for Molecular Medicine in the Helmholtz Association (MDC), Biology of Malignant Lymphomas, 13125 Berlin, Germany. [2]Hematology, Oncology, and Cancer Immunology, Charité – Universitätsmedizin Berlin, Corporate member of Freie Universität Berlin, Humboldt-Universität zu Berlin, Berlin Institute of Health, 10117 Berlin, Germany. [3]Experimental and Clinical Research Center (ECRC), a joint cooperation between Charité and MDC, Berlin, Germany. [4]Max Planck Institute of Immunobiology and Epigenetics, 79108 Freiburg, Germany. [5]Institute of Cancer and Genomic Sciences, College of Medical and Dental Sciences, University of Birmingham, Birmingham B15 2TT, UK. [6]University Medical Center Freiburg, 79106 Freiburg, Germany. [7]German Cancer Consortium (DKTK), German Cancer Research Center (DKFZ), 69120 Heidelberg, Germany. [8]Bioinformatics and Omics Data Science Platform, Berlin Institute for Medical Systems Biology, Max-Delbrück-Center, Berlin, Germany. [9]Institute of Human Genetics, Polish Academy of Sciences, Poznan 60-479, Poland. [10]Institute of Human Genetics, Christian-Albrechts-University Kiel, 24105 Kiel, Germany. [11]Centre for Molecular Medicine and Therapeutics, Department of Medical Genetics, BC Children's Hospital Research Institute, University of British Columbia, Vancouver, BC V5Z 4H4, Canada. [12]Department of Biochemistry and Pharmacology, Bio21 Molecular Science and Biotechnology Institute, The University of Melbourne, Melbourne, VIC 3000, Australia. [13]Institute of Cell Biology (Cancer Research), University of Duisburg-Essen, 45122 Essen, Germany. [14]Institute of Pathology, Universität Würzburg and Comprehensive Cancer Centre Mainfranken (CCCMF), Würzburg, Germany. [15]TRON gGmbH – Translationale Onkologie an der Universitätsmedizin der Johannes Gutenberg-Universität Mainz, Mainz, Germany. [16]Research School of Biology, The Australian National University, Canberra, ACT, Australia. [17]Max-Delbrück-Center for Molecular Medicine in the Helmholtz Association (MDC), Structural Biology, 13125 Berlin, Germany. [18]Department of Physics, Institute of Biophysics, Ulm University, Ulm, Germany. [19]Department of Physics, University of Marburg, 35052 Marburg, Germany. [20]Medical Department A for Hematology, Oncology and Pneumology, University Hospital Münster, Münster, Germany. [21]Frankfurt Institute of Advanced Studies, Frankfurt am Main, Germany. [22]Institute for Pharmacology and Toxicology, Goethe University, Frankfurt am Main, Germany. [23]Dr. Senckenberg Institute of Pathology, Goethe University Frankfurt, Frankfurt am Main, Germany. [24]Signal Transduction in Tumor Cells, Max-Delbrück-Center for Molecular Medicine, Berlin, Germany. [25]Institute for Transfusion Medicine, University of Ulm, Ulm, Germany. [26]Institute of Human Genetics, Ulm University and Ulm University Medical Center, 89081 Ulm, Germany. [27]Institute for Clinical Transfusion Medicine and Immunogenetics Ulm, German Red Cross Blood Service Baden-Württemberg-Hessen, Ulm, Germany. [28]Max-Delbrück-Center for Molecular Medicine in the Helmholtz Association (MDC), Immune Regulation and Cancer, 13125 Berlin, Germany. [29]These authors contributed equally: Nikolai Schleussner, Pierre Cauchy. ✉e-mail: stephan.mathas@charite.de

