## [Peer Review File · Nature Communications]

Transcriptional reprogramming by mutated IRF4 in lymphomaEditorial Note: Parts of this Peer Review File have been redacted as indicated to maintain the confidentiality of unpublished data.

REVIEWER COMMENTS

Reviewer #2 (Remarks to the Author):

In this manuscript, the authors present interesting findings about how a single point mutation in the DNA binding domain of the Interferon Regulatory Transcription Factor (IRF4) found in hematopoietic cancers leads to a switch between gene regulation programs by changing the DNA-binding specificity of the wild type IRF4. The mutation identified in human lymphomas is a substitution of a CYS for an ARG in the core of the DNA binding interface of IRF4 (position 99). Remarkably, this mutation diminishes the binding of IRF4 to its canonical binding motifs and enhances IRF4 binding to canonical and non-canonical composite motifs together with AP-1 (Activator Protein 1) complexes. The result of this binding mode switch is the inhibition of IRF4-dependent plasma cell induction and the activation of disease specific genes.

As my expertise is mainly in the structural biology of protein-DNA recognition, my comments below will be focused on a general assessment of the manuscript and a detailed assessment of the presented structural basis for the switch in IRF4-DNA binding properties. Therefore, my comments will not address the detailed technical aspects of the many experimental approaches used in this study.

Overall the findings presented are striking as they reveal a substantial switch in DNA recognition patterns upon a single point mutation in the DNA binding domain of a transcription factor. This is surprising because given the location of position 99 in the middle of the DNA recognition helix that docks into the major groove of the DNA, next to a positively charged R98, one would expect that the C99R mutation has a general detrimental effect on DNA binding. However, the authors present conclusive evidence that this is not the case. Although the mutation does impair the homodimer binding of IRF4 to its canonical binding motif, it does not impair the cooperativity of IRF4 with AP-1 complexes on one type of composite motif (AICE2) and the mutant is functional. The authors also identified one motif (AICE2FLIP) that is specifically recognized by the mutant IRF4-C99R. Finally, they showed that a specific T nucleotide 4 bases upstream of the sequence specific binding site which is required for the cooperative binding of the wild type IRF4 with AP-1 complexes to the AICE2 motif is not required in the case of the mutant. The authors show evidence that the different DNA binding properties of the mutant are reflected in the genomic binding as DNA motifs specifically recognized by the mutant become predominant bound in the cell lines where the mutant is expressed. These DNA binding properties translate into an altered cellular function of the mutant both in primary B cells and in cancer cells. In primary cells, it inhibits the plasma cell differentiation whereas in lymphoma cells it upregulates hallmark genes of the disease.

In general, the manuscript is well written and presents a complete overview of the effect of the C99R mutation, from DNA binding profiles to the implications for the B cell development and malignancies. It

starts with presenting the evidence for the presence of the mutation in lymphomas. Then, it presents the data showing the switch in DNA binding properties upon mutation followed by the data showing the different gene expression profiles upon inducing the WT and mutant IRF4 and the genomic binding profiles of the IRF4 variants in different cell types. The manuscript ends with the data presenting evidence for the blocking of plasma cell differentiation and the up regulation of disease specific genes by the mutant IRF4.

My major concerns regarding the manuscript are:

1. The authors only test one AP-1 heterodimer (BATF-JUN) to show the switch in the DNA binding profile of the mutant. Given that BATF factors lack one structural part comparing to other factors (e.g. FOS), (Murphy et al., Nature Rev. Immunology 2013), the findings would be stronger if they are reproducible with a different AP-1 heterodimer.

2. The lower affinity of IRF4 for the canonical motifs is attributed to an interaction between an auto-inhibitory region from the second IRF4 domain (IAD) and the DBD (Sundararaj and Casarotto, Biophys. Rev., 2021). In this manuscript, the authors attribute the change in DNA binding specificities between the wild type and mutant strictly to the DNA binding properties of the monomeric IRF4 DNA binding domain. The monomeric binding is rather irrelevant. Even on the canonical motifs, IRF4 forms homodimers inducing some changes in the DNA structure (PDBID 7JM4). The authors used neither experimental, nor structural modeling approaches to clarify how C99R affect or is affected by this auto-inhibitory mechanism. Moreover, since the binding of IRF-C99R together with AP-1 heterodimers occurs only on the AICE2 motif which has the individual binding sites juxtaposed, it remains unclear to what extent a direct interaction between AP-1 and IRF4 is responsible for the effects observed.

3. While the experiments appear very thoroughly performed and presented, the structural modeling data used to infer a structural mechanism for the switch in DNA binding preferences of IRF4-C99R is less solid and quite carelessly presented. In Figure 1 g,h,i it is not clear how these complexes were obtained and how the authors validate these models. In particular the DNA conformations illustrated especially in h and i appear unreasonably distorted. The authors do not specify if they performed flexible docking and if yes, what protocol they used. Due to the low quality of the illustrations it is not possible to read which bases are making interactions with the DBD in the three cases. The labels on the DNA are very small and there is no indication about the correspondence with the sequences presented below the structures. Some rather low quality illustrations are in expanded data figure 4. For example, it is completely unclear what is superimposed in panel a. Are these docking poses of the WT and C99R or is the structure of the WT and a docking pose of the C99R ? The methods part for the structural modeling is also very carelessly written. Because of this it remains unclear what the authors have actually done. For example what does the phrase "further validation was done by docking with all outcome scores tabulated in Extended Data Table 4" actually mean ? And there are other example of very poor writing in this section (e.g. "Initial DNA sequence models were generated using ISRE-DNA (PDB 7JM4) template based free annealing and

ternary structure prediction using HNADOCKDNA program”). The authors state “All docking studies were performed in the presence of all hydrogen atoms and water molecules (at least 5 Å around the DNA) using the Mastero package program”. Which water molecules they refer to ? Where do these waters come from ? How do they influence the docking results ?. First the authors state they performed the docking with Haddock, then with “Mastero” program (probably meaning “Maestro”) but the reference 57 points to a molecular dynamics algorithm implemented in the NAMD program. Does Maestro provide an interface to Haddock ? Or did they use different docking protocols ? Did the authors perform any MD simulations ? All these questions remain unanswered. Therefore, it is very unclear what software and algorithms were used. Further, the authors write “analysis of all individual models in each cluster”. Did they perform clustering of the docking results ? If yes, what kind of clustering and why analyzing all individual models in each cluster and not the cluster representatives ? Given these issues, I consider that in the form they are presented, the structural modeling results are not reliable and cannot be used to infer any mechanistic insights.

Reviewer #3 (Remarks to the Author):

In the present manuscript Schlessner et al. describe the novel recurrent somatic mutation c.297T>C (C99R) in IRF4 gene, located at DNA-binding domain, that is found in primary mediastinal B cell lymphomas and classical Hodgkin lymphomas. They use this novel mutation to understand further how TF-mutations impact on their DNA binding - specially to composite elements (CE) - and influence their interaction with other TFs. Through a comprehensive study they characterized that C99R mutation alters IRF4 DNA-binding capacity in HL cell lines, changing its preference to bind to non-canonical composite elements. They show that IRF4-C99R preferentially binds to AP1-IRF-CE dependent sites and that it severely blocks IRF4-dependent plasma cell induction through the dysregulation of few important plasma-cell genes. The manuscript is well conducted and convincing, and I only have minor comments.

1. The introduction and the discussion are quite short and adding more information to put it in a broader context will be highly recommended. This may include a brief explanation about composite elements importance in TFs binding/regulation, as well as the importance of IRF4 in both plasma-cell differentiation and lymphomas. The discussion may also expand a little bit more on the different results. Importantly, it will be recommended to elaborate a bit further on the inhibitors mentioned at the very end.

2. In the abstract and at the discussion the authors suggest this mutation is “disease-causing”, but the data presented here only suggest it is associated to disease: there is no real probe of causality on their work. It is recommended to use “disease-associated mutation”.

3. The authors show that IRF4-C99R mutation is able to rescue HRS cells as efficiently as IRF4-WT from cell death induced by shRNA mediated knock-down of endogenous IRF4. Did they find any differences that support its role in the plasma-cell differentiation blockage?

4. It is not well explained why the IRF4-R98A C99A mutation is non-functional. Please explain this further.

5. It is very intriguing the strong blockage of plasma cell differentiation found upon IRF4-C99R overexpression, that is even higher than the one induced by the double mutation R98A C99A. Please comment further on this finding.

Reviewer #4 (Remarks to the Author):

In this manuscript, Schleussner and colleagues focused on IRF4 C99R mutation which targets the DNA-binding domain. They showed that IRF4 C99R modulates its DNA-binding capacity, with loss-of-binding to canonical IRF motifs and neomorphic gain-of-binding to canonical and non-canonical IRF composite elements using DNase-seq, ChIP-seq, gene expression profiling, and EMSA. In addition, the authors revealed that IRF4 C99R overexpression blocks plasma cell induction, compared with IRF4 WT. Although the authors clearly demonstrated the functional impact of IRF4 C99R mutation, a recent elegant paper has already demonstrated similar results investigating an IRF4 mutant (T95R) affecting the DNA-binding domain with human samples, knock-in mice, and molecular experiments (PMID: 36662884). In addition, IRF4 mutations, including C99R mutation, have been reported in various subtypes of mature B-cell neoplasms (PMID: 29641966). Therefore, the paper seems lacking novelty.

Major comments

(1) The functional impact of IRF4 mutations affecting the DNA-binding domain has already been reported (PMID: 36662884). This paper reported a similar IRF4 mutation (T95R) identified in patients with autosomal dominant combined immunodeficiency. They beautifully showed similar results using human samples, knock-in mice, and molecular experiments. Therefore, the authors should reconsider the novelty of the paper.

(2) IRF4 mutations have been reported in various subtypes of mature B-cell neoplasms. For example, approximately 9% of cases have IRF4 mutations, including C99R mutation, in diffuse large B-cell lymphoma (DLBCL) (PMID: 29641966). Therefore, the authors should appropriately cite these papers.

(3) A number of genes, including IRF4, have been shown to be recurrent targets of aberrant somatic hypermutations (PMID: 23131835). Thus, the hotspot formation at C99R does not necessarily suggest its gain-of-function nature. Indeed, IRF4 mutations observed in DLBCL include loss-of-function mutations, such as nonsense and frameshift mutations, suggesting loss-of-function of IRF4 mutations. Therefore, although the authors clearly showed that IRF4 C99R mutations partly act in a gain-of-function manner, the more important functional consequence of IRF4 mutations seem to be loss-of-function from the genetic perspective.

(4) To show the differences between IRF4 WT and IRF4 C99R, the authors compared L428 cells harboring IRF4 C99R and KM-H2 harboring IRF4 WT in many experiments. The results may reflect the differences between IRF4 WT and IRF4 C99R, but may be affected by other biological differences between two cell lines. Therefore, it would be better to perform the same experiments comparing IRF4 WT- and IRF4 C99R-transduced cell lines or ideally CRISPR-edited isogenic cell lines.

(5) The authors argued that IRF4 C99R alters gene expression associated with plasmacytic differentiation and Hodgkin lymphoma. However, it looks like no statistical analysis was performed to show these associations (Fig 3f, 4a, and 4e). In addition, it is unclear how many and which cell lines were compared.

(6) Besides the above, the number of replicates and statistical methods used are not clear in several figures (for example Fig 3a, Fig 4b, Extended Data Fig 3, and Extended Data Fig 5).

RESPONSE TO REVIEWERS' COMMENTS

We would like to thank the reviewers for their constructive comments and suggestions. As outlined below, we have addressed all points raised by the referees. The repeated or paraphrased comments of the reviewers are shown in *italics*. All changes in the manuscript are marked-up in the respective marked-up version of the manuscript (underlined and highlighted in yellow).

Reviewer #2

“In this manuscript, the authors present interesting findings about how a single point mutation in the DNA binding domain of the Interferon Regulatory Transcription Factor (IRF4) found in hematopoietic cancers leads to a switch between gene regulation programs by changing the DNA-binding specificity of the wild type IRF4. The mutation identified in human lymphomas is a substitution of a CYS for an ARG in the core of the DNA binding interface of IRF4 (position 99). Remarkably, this mutation diminishes the binding of IRF4 to its canonical binding motifs and enhances IRF4 binding to canonical and non-canonical composite motifs together with AP-1 (Activator Protein 1) complexes. The result of this binding mode switch is the inhibition of IRF4-dependent plasma cell induction and the activation of disease-specific genes.

As my expertise is mainly structural biology of protein-DNA recognition, my comments below will be focused on a general assessment of the manuscript and a detailed assessment of the presented structural basis for the switch in IRF4-DNA binding properties. Therefore, my comments will not address the detailed technical aspects of the many experimental approaches used in this study.

Overall the findings presented are striking as they reveal a substantial switch in DNA recognition patterns upon a single point mutation in the DNA binding domain of a transcription factor. This is surprising because given the location at position 99 in the middle of the DNA recognition helix that docks into the major groove of the DNA, next to a positively charged R98, one would expect that the C99R mutation has a general detrimental effect on DNA binding. However, the authors present conclusive evidence that this is not the case. Although the mutation does impair the homodimer binding of IRF4 to its canonical binding motif, it does not impair the cooperativity of IRF4 with AP-1 complexes on one type of composite motif (AICE2) and the mutant is functional. The authors also identified one motif (AICEFLIP) that is specifically recognized by the mutant IRF4-C99R. Finally, they showed that a specific T nucleotide 4 bases upstream of the sequence specific binding site which is required for the cooperative binding of the wild type IRF4 with AP-1 complexes to the AICE2 motif is not required in the case of the mutant. The authors show evidence that the different DNA binding properties of the mutant are reflected in the genomic binding as DNA motifs specifically recognized by the mutant become predominant bound in the cell lines where the mutant is expressed. These DNA binding properties translate into an altered cellular function of the mutant both in primary B cells and in cancer cells. In primary cells, it inhibits the plasma cell differentiation whereas in lymphoma cells it upregulates hallmark genes of the disease.

In general, the manuscript is well written and presents a complete overview of the effect of the C99R mutation, from DNA binding profiles to the implications for the B cell development and malignancies. It starts with presenting evidence for the presence of the mutation in lymphomas. Then, it presents the data showing the switch in DNA binding properties upon mutation followed by the data showing the different gene expression profiles upon inducing the WT and mutant IRF4 and the genomic binding profiles of the IRF4 variants in different cell types. The manuscript ends with the data presenting evidence for the blocking of plasma cell differentiation and the up regulation of disease specific genes by the mutant IRF4.”

We sincerely thank the Reviewer for the overall very positive comments.

“My major concerns regarding the manuscript are:

1. The authors only test one AP-1 heterodimer (BATF-JUN) to show the switch in the DNA binding profile of the mutant, Given that BATF factors lack one structural part comparing to other factors (e.g. FOS), (Murphy et al., Nature Rev. Immunology 2013), the findings would be stronger if they are reproducible with a different AP-1 heterodimer.”

We thank the reviewer for this comment. We performed our experiments with AP-1 heterodimers consisting of BATF-JUNB for the following reasons:

- a) These dimers were identified as the most relevant complexes at AICEs (e.g. Glasmacher *et al.*, Science, 2012, PMID 22983707). However, apart from JUNB, it is known that also JUN-BATF AP-1 heterodimers form IRF4 composite complexes at AICEs (Glasmacher *et al.*, Science, 2012, PMID 22983707; Tussiwand *et al.*, Nature, 2012, PMID 22992524). In contrast, neither FOS nor its family member FOSL2 can form IRF4 composite complexes at AICEs (Glasmacher *et al.*, Science, 2012; Tussiwand *et al.*, Nature, 2012).
- b) We and others have previously demonstrated that Hodgkin lymphoma cells are characterized by a constitutive activity and high-level expression of the AP-1 family members JUNB, JUN, BATF and BATF3 (e.g. Mathas *et al.*, EMBO J, 2002, PMID 12145210; Schleussner *et al.*, Leukemia, 2018, 29588546; Lollies *et al.*, Leukemia, 2018, 28659618), whereas FOS is not found within the constitutively activated AP-1 complex in HRS cells.

To answer the reviewer's comment we now include the following new data in the revised version of our manuscript (new Extended Data Figures 3e and 3f and Extended Data Figures 12b and 12c):

- a) We performed additional DNA-binding studies with c-JUN instead of JUNB together with BATF and IRF4 at sites AICE1 (Ctla4), AICE1 (IL12Rb), AICE2 (BCL11b) and AICE2 -4C (Bcl11b) – new Extended Data Figures 3e and 3f.
- b) We also performed DNA-binding studies with BATF3 instead of BATF together with JUNB and IRF4 at sites *GATA3Peak_1* and *GATA3Peak_2* – new Extended Data Figures 12b and 12c.

These new experiments show similar to those with JUNB. They confirm our previous findings on the switch of the mutant IRF4-C99R protein with respect to its DNA-binding properties and demonstrate that this switch is not restricted to its interaction with JUNB-BATF heterodimers. We modified the Result and Discussion section of the revised version of our manuscript accordingly.

“2. The lower affinity of IRF4 for the canonical motifs is attributed to an interaction between an auto-inhibitory region from the second IRF4 domain (IAD) and the DBD (Sundararaj and Casarotto, Biophys Rev., 2021). In this manuscript, the authors attribute the change in DNA binding specificities between the wild type and mutant strictly to the DNA binding properties of the monomeric IRF4 DNA binding domain. The monomeric binding is rather irrelevant. Even on the canonical motifs, IRF4 forms homodimers including some changes in the DNA structure (PDBID 7JM4). The authors used neither experimental, nor structural modeling approaches to clarify how C99R affect or is affected by this auto-inhibitory mechanism. Moreover, since the binding of IRF4-C99R together with AP-1 heterodimers occurs only in the AICE2 motif which

has the individual binding sites juxtaposed, it remains unclear to what extent a direct interaction between AP-1 and IRF4 is responsible for the effects observed.”

We thank the reviewer for this insightful comment. In fact, the low affinity binding of IRF4 to DNA has been attributed to its auto-inhibitory region. It has been proposed that in case of cooperative binding, autoinhibition is relieved, thus facilitating IRF4 DNA-binding e.g. to composite elements (CEs). However, whilst IRF4 indeed binds as homodimeric complex to DNA at canonical ISRE motifs consisting of multiple 5′ - GAAA - 3′ IRF consensus motifs, the situation at EICE or AICE composite elements (CEs) is different. In case of EICE, one single IRF4 molecule binds together with PU.1 to form heterodimeric complexes (see e.g. crystal structure-analysis in Escalante *et al.*, Mol Cell, 2002, PMID 12453417), in case of AICE one single IRF4 binds together with two different AP-1 factors to form heterotrimeric complexes (see molecular modeling in Glasmacher *et al.*, Science, 2012, PMID 22983707). Regarding AICE, the exact cooperative binding mechanism, and which protein regions/residues exactly mediate cooperative binding between IRF4 and AP-1 TFs have not yet been clarified in detail (Tussiwand *et al.*, Nature, 2012, PMID 22992524; Murphy *et al.*, Nat Rev Immunol, 2013, PMID 23787991). In our work, we have demonstrated that IRF4-C99R exerts a remarkable shift in DNA-binding specificity, and that binding at specific AICEs is largely dependent on its cooperative activity with AP-1 TFs, as demonstrated for example in Figures 1d, 1e, 1f, 4b. Furthermore, we performed experiments with only the DNA-binding domain (DBD) of IRF4-WT and IRF4-C99R lacking the autoinhibitory domain (Extended data Figures 5b and 5c), which fully confirmed and mirrored our findings with full-length IRF4 proteins. Together with our structure modelling, these data suggest that C99R primarily modulates the intrinsic properties of the IRF4 DNA-binding domain to recognize specific DNA motifs. We agree with the reviewer that the question whether direct interaction between IRF4 and AP-1 TFs is contributing to these effects has not yet been clarified in detail. However, we believe that our manuscript carries enough weight to inspire studies of this interesting question for future studies.

We modified the Discussion section of our revised manuscript accordingly.

“3. While the experiments appear very thoroughly performed and presented, the structural modeling data used to infer a structural mechanism for the switch in DNA binding preferences of IRF4-C99R is less solid and quite carelessly presented. In Figure 1 g,h,i it is not clear how these complexes were obtained and how the authors validate these models.

This part of the reviewer’s comment refers to the starting template used, the model validation and the clustering of low energy structures. Foremost, the structural modeling is aimed at providing additional information that informs how the IRF4-C99R mutation influences DNA-interaction compared to the WT and at the different DNA-binding motifs addressed

experimentally. As mentioned in our methods section (page 39, lines 806-807), for all structural models, the initial model structures of IRF4 and DNA were obtained from our previous crystal structure (PDB:7JM4). As only one IRF4 molecule binds to AICEs, we incorporated only one in the modeled structures. As shown in Extended data Table 4, the most appropriate models were considered based on resultant docking parameters such as HADDOCK score, cluster size, or desolvation energy. The Methods and Result sections have been modified accordingly in the revised version of our manuscript to further clarify what we have done.

In particular the DNA conformations illustrated especially in h and i appear unreasonable distorted.

In our view, this point of the reviewer particularly refers to Fig. 1h, left, which is meant to show an inability of the mutant protein to bind to the AICE1 sequence. Due to the low T_m and poor annealing in the modelled structure, the DNA conformation appears more distorted – in essence the computational work tries its best to fit the two molecules together. However, the low HADDOCK score, the high RMSD and high Z-score (refer to Extended Data Table 4) indicated either no interaction or a poor model. By asking flexible docking methods to come up with an unrefined model, the quality of the model is indeed poor. In summary, modelling shows that C99R does not bind to AICE1, which is consistent with our EMSA results. We could remove this panel from Figure 1. However, keeping this panel truly visualizes the differences between IRF4-C99R interactions with AICE1 or AICE2, as observed in our DNA-binding studies by EMSA. In the revised version of our manuscript, we preferred the latter option and have added additional explanations in the Result section of the revised version of our manuscript.

The authors do not specify if they performed flexible docking and if yes, what protocol they used.

In the HADDOCK program we mainly used the flexible docking approaches for modelling the biomolecular processes. To drive the docking process, the HADDOCK programme differs from ab-initio docking methods by utilizing information from known or predicted protein interfaces in ambiguous interaction restraints (AIRs). We have now included this information in the Methods section.

Due to the low quality of the illustrations it is not possible to read which bases are making interactions with the DBD in the three cases. The labels on the DNA are very small and there is no indication about the correspondence with the sequences presented below the structures.

We thank the reviewer for this comment; the quality all the figure parts has been improved in the revised versions of Figures 1g-h and Extended Data Figure 4.

Same rather low quality illustrations are in expanded data figures 4. For example, it is completely unclear what is superimposed in panel a. Are these docking poses of the WT and C99R or is the structure of the WT and a docking pose of the C99R?

We took this comment to heart and have improved the illustrations in Extended data Figure 4. In Extended Data Figure 4a, the IRF4-WT/ISRE DNA complex (PDB:7JM4) is super-imposed with the modelled C99R/ISRE DNA complex (colour code for DNA and IRF4 were retained). The change or mutation C99R is shown as a stick image. IRF4-C99R could not interact with the intact ISRE DNA (see Fig. 1g), and in order to accommodate the DNA interaction, C99R needs to adjust the positioning of the ISRE DNA (shown here). The unbiased and reference free structural modelling of IRF4-C99R with the ISRE DNA shows bending of the DNA and displacement of the phosphate backbone to accommodate the recognition helix into the major groove of the DNA. Without this structural re-arrangement of ISRE DNA, binding of IRF4-C99R is difficult. We have now outlined these issues in the Result section of our revised manuscript.

The methods part for the structural modeling is also very carelessly written. Because of this it remains unclear what the authors have actually done. For example what does the phrase “further validation was done by docking with all outcome scores tabulated in Extended Data Table 4” actually mean?

To corroborate the modeled structures, several modeling outcome parameters were analyzed according to HADDOCK guidelines (<https://www.bonvinlab.org/software/haddock2.2/analysis/>) as tabulated in the Extended Data Table 4. Primarily the higher negative values of HADDOCK score, Z-score, van der Waals and electrostatic energy were considered to assess the better models. In all parameters, lower values denote better models as opposed to the buried interface and desolvation energies were higher values signify greater confidence in the binding model. We modified the Methods section of the revised version of our manuscript accordingly.

And there are other example of very poor writing in this section (e.g. “Initial DNA sequence models were generated using ISRE-DNA (PDB 7JM4) template based free annealing and tertiary structure prediction using HNADOCKDNA program”). The authors state “All docking studies were performed in the presence of all hydrogen atoms and water molecules (at least 5Å around the DNA) using the Mastero package program.” Which water molecules they refer to? Where do these waters come from? How do they influence the docking results? First the authors state they performed the docking with Haddock, then with “Mastero” program (probably meaning “Maestro”) but the reference 57 points to a molecular dynamics algorithm

implemented in the NAMD program. Does Maestro provide an interface to Haddock? Or did they use different docking protocols? Did the authors perform any MD simulations? All these questions remain unanswered. Therefore, it is very unclear what software and algorithms were used. Further, the authors write “analysis of all individual models in each cluster”. Did they perform clustering of the docking results? If yes, what kind of clustering and why analyzing all individual models in each cluster and not the cluster representatives? Given these issues, I consider that in the form they are presented, the structural modeling results are not reliable and cannot be used to infer any mechanistic insights.”

We thank the reviewer for bringing this to our attention. As suggested, in the revised manuscript we now have rewritten the methods part to provide a clearer workflow. We have to admit that mentioning the Maestro program was a mistake in the initial version of our manuscript, and it has been actually corrected.

In some programs, use of hydrogens is optional in the PDB structure and is recommended for better results. We used all the hydrogen atoms present in the PDB structure for docking studies. With regard to water molecules, the HADDOCK program has an option to perform the docking studies virtually adding water molecules to any water-mediated solvation contacts. In our studies with IRF4 and DNA elements we noticed little difference in the results with the addition or absence of virtual water molecules. The HNADOCKDNA program (part of the HADDOCK server) was used only to anneal two different ssDNA elements but was not used for the docking of IRF4. Furthermore, the clustering part has been reworded for more clarity in the revised Methods section of our manuscript.

Reviewer #3

“In the present manuscript Schleussner et al. describe the novel recurrent somatic mutation c.297T>C (C99R) in IRF4 gene, located at DNA-binding domain, that is found in primary mediastinal B cell lymphomas and classical Hodgkin lymphomas. They use this novel mutation to understand further how TF-mutations impact on their DNA binding – specially to composite elements (CE) – and influence their interaction with other TFs. Through a comprehensive study they characterized that C99R mutation alters IRF4 DNA-binding capacity in HL cell lines, changing its preference to bind to non-canonical composite elements. They show that IRF4-C99R preferentially binds to API-IRF-CE dependent sites and that it severely blocks IRF4-dependent plasma cell induction through the dysregulation of few important plasma-cell genes. The manuscript is well conducted and convincing, and I only have minor comments.”

We sincerely thank the Reviewer for his/her very positive comments.

Minor comments:

“1. The introduction and the discussion are quite short and adding more information to put in a broader context will be highly recommended. This may include a brief explanation about composite elements importance in TFs binding /regulation, as well as the importance of IRF4 in both plasma-cell differentiation and lymphomas. The discussion may also expand a little bit more on the different results, importantly, it will be recommended to elaborate a bit further on the inhibitors mentioned at the very end.”

We thank the reviewer for these suggestions. We modified the Introduction and the Discussion section of our manuscript accordingly.

“2. In the abstract and at the discussion the authors suggest this mutation is “disease-causing”, but the data presented here only suggest it is associated to the disease: there is no real probe of causality on their work. It is recommended to use “disease-associated mutation.”

To follow the suggestion of the reviewer, we prefer to take out the wording “disease-causing” at two places in the revised abstract and discussion sections of our manuscript. Instead of “...to block the neomorphic, disease-causing DNA binding activities of a mutant transcription factor.” this phrase now reads “...to block the neomorphic DNA binding activities of a mutant transcription factor.”

“3. The authors show that IRF4-C99R mutation is able to rescue HRS cells as efficiently as IRF4-WT from cell death induced by siRNA mediated knock-down of endogenous IRF4. Did they found any differences that support its role in the plasma-cell differentiation blockage?”

We thank the reviewer for this comment. According to the current concept of cHL pathogenesis, HRS cells originate in most cases from (post-)Germinal Center (GC) B cells and show features of abortive plasma cell differentiation. The formation of the block in terminal B cell differentiation in HRS cells is complex due to multiple complementary mechanisms. Thus, for example, functional inhibition of key lineage-specific and lineage-instructive transcription factors such as E2A, genetic alterations of key transcriptional regulators such as PU.1, lineage-inappropriate gene expression, or epigenetic alterations contribute to the disrupted terminal B cell differentiation of HRS cells (Küppers R, Nat Rev Cancer, 2009, PMID 19078975; Mathas *et al.*, Nat Immunol, 2006, PMID 16369535; Lamprecht *et al.*, Nat Med, 2010, PMID 20436485; Ehlers *et al.*, Leukemia, 2018, PMID 18256685; Seitz *et al.*, Haematologica, 2011, PMID 21393330; Ushmorov *et al.*, Blood, 2006, PMID 16304050). To overcome this block and to restore terminal plasma cell differentiation is challenging, and even if some B cell-specific or plasma cell-associated genes could be reactivated in HRS cells in some studies, usually protein expression of the respective genes was not detectable (see e.g. Bohle *et al.*, Leukemia, 2013, PMID 23174882; Osswald *et al.*, Blood, 2018, 29439954; Hertel *et al.*, Oncogene, 2002, PMID

12118370). In the case of IRF4-C99R, we did not analyze gene expression changes in the mentioned experiment, which however is an interesting question for future studies. In the revised version of our manuscript, we now modified the Discussion section of our manuscript to address the point raised by the reviewer.

“4. It is not well explained why the IRF4-R98A C99A mutation is non-functional. Please explain this further.”

With one exception, positions corresponding to R98 and C99 in IRF4 are conserved across all IRF family members (see Extended data Fig. 1e). It is well established that both residues are critically involved in the formation of IRF:DNA complexes (e.g. Escalante *et al.*, Nature, 1998, PMID 9422515; Escalante *et al.*, Mol Cell, 2002, PMID 12453417), and that mutation of these residues abolishes IRF(4) DNA-binding and function (e.g. Brass *et al.*, EMBO J, 1999, PMID 10022840; Sciammas *et al.*, Immunity, 2006, PMID 16919487). To follow the suggestion of the reviewer, we now include these citations in the revised version of our MS and modified the Result section accordingly.

“5. It is very intriguing the strong blockage of plasma cell differentiation found upon IRF4-C99R overexpression, that is even higher than the one induced by the double mutation R98A C99A. Please comment further on this finding.”

It is indeed a remarkable finding that the blockage of plasmablast formation by ectopic IRF4-C99R expression is even higher than that of the double mutant R98AC99A, which demonstrates the dominant effect of the mutant IRF4-C99R protein. In the new version of Fig. 3a, right, the respective statistic has now been added. Possible explanations might be that IRF4-C99R-regulated genes interfere with plasmablast formation, or that IRF4-C99R might form heterodimeric complexes with IRF4-WT unable to bind to DNA, which remains to be evaluated in future studies. We now modified the Discussion section of the revised version of our manuscript accordingly.

Reviewer #4

“In this manuscript, Schleussner and colleagues focussed on IRF4 C99R mutation which targets the DNA-binding domain. They showed that IRF4 C99R modulates its DNA-binding capacity, with loss-of-binding to canonical IRF motifs and neomorphic gain-of-binding to canonical and non-canonical IRF composite elements using DNase-seq, ChIP-seq, gene expression profiling, and EMSA. In addition, the authors revealed that IRF4 C99R overexpression blocks plasma cell induction, compared with IRF4 WT. Although the authors clearly demonstrated the functional impact of IRF4 C99R mutation, a recent elegant paper has already demonstrated similar results investigating an IRF4 mutant (T95R) affecting the DNA-binding domain with human samples,

knock-in-mice, and molecular experiments (PMID: 36662884). In addition, IRF4 mutations, including C99R mutation, have been reported in various subtypes of mature B-cell neoplasms (PMID: 29641966). Therefore, the paper seems lacking novelty.”

Major concerns:

“(1) The functional impact of IRF4 mutations affecting the DNA-binding domain has already been reported (PMID: 36662884). This paper reported a similar mutations (T95R) identified in patients with autosomal dominant combined immunodeficiency. They beautifully showed similar results using human samples, knock-in-mice, and molecular experiments. Therefore, the authors should reconsider the novelty of the paper.”

We are fully familiar with the mentioned publication by Fornes *et al.* (PMID 36662884) describing and characterizing the IRF4-T95R mutation in combined immunodeficiency (CID). Note that the first and last author of the current C99R manuscript are among the group of first and last authors on the T95R paper. Moreover, for full wtransparency, the T59R manuscript had been announced in the cover letter and was fully provided during the review process of our C99R MS.

We strongly disagree with the statement of the reviewer that the publication of the T95R mutation in CID leads to a lack of novelty of our C99R data. We believe that both papers are highly complementary and highlight the relevance of our findings for lymphoid biology and disease pathogenesis in general. More specifically, while T95R is a germline mutation, C99R is a somatic one. T95R results in combined immunodeficiency with so far no evidence for an impact on malignant transformation of lymphoid cells, but C99R is associated with human lymphomas characterized by perturbed B cell identity. For both mutations, we provide evidence that the respective mutated IRF4 proteins have lost the ability to regulate canonical IRF4 target genes and are unable to exert its physiological function to coordinate plasma cell differentiation, whereas they instead upregulate disease-associated genes which cannot be regulated by IRF4-WT. However, at the level of DNA-binding, both mutations remarkably differ. Overall, T95R shows a broadly increased DNA-binding affinity to canonical and non-canonical DNA motifs, even though T95R motif-preference is shifted compared to IRF4-WT (Fornes *et al.*, PMID 36662884). In our current manuscript, we provide highly convincing molecular evidence that C99R has a different effect on DNA-binding with an unprecedented combination of loss-gain pattern, and, overall, effects on DNA-binding properties are more dramatic with respect to motif recognition compared to T95R. Together, both IRF4 mutations point to the complex shift in transcription factor DNA-binding specificity and gene regulation in different diseases induced by a single arginine mutation within the DNA-binding domain. We now modified the discussion section of our manuscript accordingly.

“(2) IRF4 mutations have been reported in various subtypes of mature B-cell neoplasms. For example, approximately 9% of cases have IRF4 mutations, including C99R mutation, in diffuse large B-cell lymphoma (DLBCL) (PMID: 29641966). Therefore, the authors should appropriately cite these papers.”

We thank the reviewer for this comment - we are fully aware that IRF4 mutations including C99R have been reported in lymphoid malignancies including mature B cell lymphomas (see initial version of our manuscript, in which we included data from COSMIC; Extended data Fig. 1b, lower part). However, if counting somatic variants, those introduced by aberrant somatic hypermutation (affecting the first 2-2.5kb genomic sequence from transcription initiation site (TIS) and thus in the 5' part of the gene including (non-coding) exon 1 and exon 2) have to be distinguished from those mutations located 3', i.e. from exon 3 to the 3' UTR. The vast majority of the 9% of mutations quoted by the reviewer belong to the prior group in the 5' part of the gene, whereas the C99R located in exon 3 and, thus, >3kb downstream of the TIS belongs to the much rarer more 3' mutations. Indeed, we already stated in the legend to Extended data Fig. 1 that, of note, among the 9 samples harboring C99R in COSMIC, three samples are annotated as PMBCL and one as cHL. Revisiting the original publications from the remaining 5 samples with IRF4-C99R reported in COSMIC, at least one DLBCL sample is a PMBCL (Mareschal *et al.*, Genes Chromosomes Cancer, 2016; PMID 26608593). Thus, IRF4-C99R is reported in 3 of 2363 samples other than cHL or PMBCL, which is far below the frequency we report for cHL and PMBCL and supports our notion that IRF4-C99R is an exceptional finding in lymphoid malignancies other than cHL and PMBCL.

In addition, we revisited the data from three recent papers reporting on genetics of large DLBCL cohorts (Reddy *et al.*, Cell, 2017, PMID 28985567; Chapuy *et al.*, Nat Med., 2018, PMID 29713087; Schmitz *et al.*, N Engl J Med, 2018, PMID 29641966). Overall, whereas various IRF4 mutations are reported in DLBCL, IRF4-C99R detection is exceptional. Thus, as far as the data are extractable from the papers, among a total of 1742 DLBCL cases a maximum of 6 cases with IRF4-C99R is reported (0,29 %).

Due to space limitations we did not cite individual papers reporting on C99R in lymphoid malignancies other than cHL and PMBCL in the initial version of our manuscript. Now, we include these citations in the Result section of our revised manuscript.

“(3) A number of genes, including IRF4, have been shown to be recurrent targets of aberrant somatic hypermutations (PMID: 23131835). Thus, the hotspot formation at C99R does not necessarily suggest its gain-of-function nature. Indeed, IRF4 mutations observed in DLBCL include loss-of-function mutations, such as nonsense and frameshift mutations, suggesting loss-of-function of IRF4 mutations. Therefore, although the authors clearly demonstrate that IRF4 C99R mutations partly act in a gain-of-function manner, the more important functional consequence of IRF4 mutations seem to be loss-of-function from the genetic perspective.”

We thank the Reviewer for this insightful comment. As already mentioned above (see our response to point (2) of this reviewer), various genes including *IRF4* have indeed been shown to be targets of aberrant somatic hypermutation (SHM) in lymphoid malignancies including DLBCL (see e.g. Pasqualucci *et al.*, *Nature*, 2001, PMID 11460166; Khodabakhshi *et al.*, *Oncotarget*, 2012, PMID 23131835). Typically, aberrant hypermutation activity affects regions spanning about 2-2.5kb downstream from the transcription initiation site (TIS) of the respective gene (see e.g. Storb *et al.*, *Immunol Rev.*, 1998, PMID 9602361; Pasqualucci *et al.*, *Nature*, 2001, PMID 11460166). To further analyze whether this pattern also holds true for *IRF4* mutations in B-cell lymphomas, we analyzed the distribution of mutations in 9 *IRF4* rearranged large B-cell lymphomas from the ICGC MMML-Seq network. In these lymphomas, aberrant somatic hypermutation of the *IRF4* locus is particularly fostered by juxtaposition to the *IGH* locus. As can be seen in the Figure 1 for review only, the mutations indeed cluster in 2-2.5kb from TIS (blue bar) in *IRF4*, whereas exon 3 containing C99 (red arrow) is not affected.

Thus, C99R located >3kb downstream of the TIS in Exon 3 is not in the region undergoing somatic hypermutation in B-cell lymphomas, and thus we considered aberrant hypermutation not as causative for C99R formation. In support of this view, the nucleotide sequence around C99R does not contain the hotspot motif RGYW for SHM. Together, these data support our view that C99R is not caused by aberrant SHM.

Regarding the functional consequence of the C99R mutant, our data clearly demonstrate that this mutant exerts gain-of-function properties. Undoubtedly, the C99R mutant shows loss-of-function properties, which might be equally important for lymphoma biology (e.g. block in differentiation). We now modified the Result and the Discussion section of our revised manuscript accordingly to address for the reviewer's comment.

“(4) To show the differences between IRF4 WT and IRF4 C99R, the authors compared L428 cells harbouring IRF4 C99R and KM-H2 harboring IRF4 WT in many experiments. The results may reflect the differences between IRF4 WT and IRF4 C99R, but may be affected by other biological differences between two cell lines. Therefore, it would be better to perform the same experiments comparing IRF4 WT- and IRF4 C99R-transduced cell lines or ideally CRISPR-edited isogenic cell lines.”

In our manuscript, we provide multiple layers of evidence about the different molecular properties and functions of *IRF4*-C99R compared to *IRF4*-WT. To give a brief overview, we used ectopic expression in HEK293 cells (e.g. Fig 1), Tet-inducible expression systems in non-Hodgkin lymphoma cells (Extended data Fig. 2), or transduced primary C57BL/6 mouse splenic B cells (e.g. Fig. 3). We used these cellular systems complementary to other experiments performed with the cell lines L428 harboring *IRF4*-C99R and KM-H2 harboring *IRF4*-WT (e.g. Fig. 2). Both L428 and KM-H2 are among the best-characterized Hodgkin lymphoma-derived

cell lines and have been shown in numerous publications to be reliable models of the disease. Both cell lines are of B cell origin and have been generated from advanced-stage disease patients. To more specifically answer to the reviewer's question on the biological differences (or *vice versa* similarities) between the cell lines we revisited our RNA-seq and DHS data from the various cell lines. First, Figure 2a for review only shows a Principal Component Analysis (PCA) on the RNA-seq data from the various cell lines. Clearly, all the non-Hodgkin cell lines (Reh, Namalwa, BJAB, SH-DHL-4) cluster separately from the HL cell lines. Among the latter, KM-H2 is the second closest cell line to L428, pointing to their biological similarity. Second, we performed a correlation analysis of the available DHS data from the cell lines (Figure 2b for review only). Also at the DHS level, KM-H2 is clustering as second closest cell line to L428. Together, we believe that we used in our manuscript appropriate model systems to characterize the molecular and functional differences between IRF4-C99R and IRF4-WT.

“(5) The authors argued that IRF4 C99R alters gene expression associated with plasmacytic differentiation and Hodgkin lymphoma. However, it looks like no statistical analysis was performed to show these associations (Fig 3f, 4a and 4e). In addition, it is unclear how many and which cell lines were compared.”

We thank the reviewer for this comment. The requested information has been added in the revised version of our manuscript.

“(6) Besides the above, the number of replicates and statistical methods used are not clear in several figures (for example Fig 2a, Fig 4B, Extended Data Fig 3, Extended Data Fig 5).”

We thank the reviewer for this comment. The number of replicates and statistical methods have now been added in the revised version of the manuscript.

[Redacted]

L428 and KM-H2 cell lines are closely related at the gene expression and chromatin landscape levels. (a) Principal component (PC) analysis of 8 cHL (HL, orange) and 4 non-cHL (NHL, green) cell lines. cHL and non-cHL cell lines are clearly separated and KM-H2 is the second closest cell line to L428. (b) Hierarchical clustering of Spearman correlation coefficients of genome-wide DNase-Seq signals from 4 cHL (L591, L428, KM-H2, L1236) and 2 non-cHL (REH, NAMALWA) cell lines. cHL and non-cHL cell lines are clearly separated and KM-H2 cells are the second closest to L428 cells.

REVIEWER COMMENTS

Reviewer #2 (Remarks to the Author):

The authors nicely addressed my first two major points, performing new experiments which further confirm the proposed mechanism. The authors also included more explanation about the procedure they used for the structural modeling. However, I am afraid this still lacks clarity and it is still not possible to decide whether their modeling and docking results are reliable to conclude about the differences in DNA-binding preferences for the wild type IRF4 and the C99R mutant. This despite the very nice agreement between the modeling efforts and the experimental data. The structural modeling procedure is an elegant way to explain the experimental observations but it needs to be clear and free of any potential artifacts.

In particular:

1. The DNA structures in the Figures 1g,h and Extended Data Figure 4 still look unreasonably distorted and not only at the ends but also in the central region of the DNA. For example around bases 13-15 some of the base pairing looks compromised in most panels (even in those showing competent complexes). I haven't found the structural models in the submitted data (e.g. as PDB files), so I can only comment on the figures provided.

2. The authors mention they used HNADOCK to generate the structures of the DNA elements. However, they do not explain how they did that. When looking at HNADOCK (<http://huanglab.phys.hust.edu.cn/hnadock/>), I do not see an option to generate double stranded DNA structures. The procedure needs to be explained in detail so that it can be reproduced. It might be that this procedure leads to distortions in the DNA structure. With these distortions, I am not convinced the docking results are reliable even though the results seem to fit very well with the experimental data. Are the authors able to reproduce their own crystal structure if they generate the ISRE DNA with the same protocol ?

3. In their modified text, the authors specify: "All of the docking studies and modelling were performed using standard HADDOCK 2.2 with flexible docking parameters and without any additional restrains. The HADDOCK programme differs from ab-initio docking methods by utilising information from known or predicted protein interfaces for ambiguous interaction restraints (AIRs)."

This is a very unclear statement. First, the authors write they did not use any restraints, then they specify that HADDOCK is a docking software that uses ambiguous interaction restraints. Does this mean

that the authors did not use any of these ambiguous restraints in HADDOCK ? If this is the case, can the authors reproduce their previous crystal structure with a such a protocol devoid of any restraints ?

Moreover, they mention “flexible docking” without clearly specifying the protocol. HADDOCK uses different stages of flexible docking. It is not explained which stages the authors used. Was flexible refinement done with MD simulations or only with energy minimization ? How many models were generated ? How many clusters ? The “cluster size” in Extended Data Table 4 is for which cluster (what does “best or top complex structure” refer to) ? What “RMSD” is shown in the same table (e.g. interface, entire protein, entire complex, which atoms were used for the fit ?) ? All these are questions that are still not answered.

4. It remains unclear what is superposed in Expanded Data Figure 4a (not written in the legend). Are these docked complexes of the WT and C99R mutant on the ISRE DNA ? Or is the crystal structure in the case of the WT ? The Extended Data Table 4 does not contain data for the docking of the WT to IRSE which suggests the authors did not perform this experiment. However the orange DNA looks more distorted than the crystal structure.

Perhaps the authors could consider providing all relevant structural models and the input files and parameters for the programs they used. Including also the initial DNA structures used for the docking.

Reviewer #3 (Remarks to the Author):

The authors have addressed all my initial concerns and I have no further criticisms.

Reviewer #4 (Remarks to the Author):

The authors have substantially revised and improved the manuscript based on comments from the reviewers.

RESPONSE TO REVIEWERS' COMMENTS

Response to the comments of Reviewer #2

Reviewers #3 and #4 did not have further criticisms. Reviewer #2 agreed on our response to his/her first two major points, but some issues remained to be clarified for this reviewer regarding our modelling and docking results. We would like to thank the reviewer for the constructive comments and suggestions. As outlined below, we have addressed all points raised by the referee. The repeated or paraphrased comments of the reviewer are shown in *italics*. All changes in the manuscript are marked-up in the respective marked-up version of the manuscript (underlined and highlighted in yellow).

Reviewer #2

“The authors nicely addressed my first two major points, performing new experiments which further confirm the proposed mechanism. The authors also include more explanation about the procedure they used for the structural modelling.”

We sincerely thank the Reviewer for the overall very positive comment.

“However, I am afraid this still lacks clarity and it is still not possible to decide whether their modelling and docking results are reliable to conclude about the differences in DNA-binding preferences for the wild type IRF4 and the C99R mutant. This despite the very nice agreement between the modeling efforts and the experimental data. The structural modeling procedure is

an elegant way to explain the experimental observations but it needs to be clear and free of any potential artifacts.

In particular

1. The DNA structures in the Figures 1g,h and Extended Date Figure 4 still look unreasonably distorted and not only at the ends but also in the central region of the DNA. For example around bases 13-15 some of the base pairing looks compromised in most panels (even in those showing competent complexes). I haven't found the structural models in the submitted data (e.g. as PDB files), so I can only comment on the figures provided."

In response to the reviewer's comment on the distortion of the DNA phosphate backbone, we would like to provide some clarification. Firstly, we want to emphasize that Figure 1g in our study is derived solely from the crystal structure (PDB 7JM4) published in NAR 2021 (PMID 33533913). As now clearly outlined in the figure legend and main text of our manuscript, we have demonstrated that simply substituting C99 with R in IRF4 without any other structural alterations results in a clash surrounding R99. Just to reiterate, Figure 1g and the PDB 7JM4 both depict the same DNA structure without any changes.

Secondly, the observation of a non-uniform DNA phosphate backbone has been accentuated by how we have selected to represent the DNA phosphate backbone. In this case, we have used a thin line rather than a thicker line or bulky flat ribbon model commonly seen in textbooks of general figures. We have chosen a thin line so as not to obscure the DNA base pairs.

Moreover, the fact that the reviewer considers the experimental X-ray structure as having distorted DNA backbone weakens the argument that the DNA distortions are as a result of unreliable modelling results.

Regarding Figure 1h, left, we acknowledge that the DNA structure is more distorted than the other structures. We have been clear within our manuscript that we were unable to generate a high-quality, energy-minimized model for the IRF4-C99R:AICE1 interaction (BP1; Figure 1h, left), which is indicative of a lack of robust DNA interaction of IRF4-C99R with AICE1 in the modelling, in line with our experimental data. The modelled complex exhibited low confidence, did not show significant binding or interaction (high Z-score, refer to Extended Date Table 4) and was judged as a low quality complex structure. In combination with data in Extended Data Table 4, the poor model shown in Figure 1h, left, highlights the difference between high- and low-quality models, i.e. models showing robust interaction with their respective DNA structures (here IRF4-C99R:AICE2 model; now Figure 1i) and those not doing so (here IRF4-C99R:AICE1 model; now Figure 1h). For this reason, we decided to keep Figure

1h, left, within our figure panels. For clarity, the panel of Figure 1h has been restructured into Figures 1h and 1i, and the respective Figure legend has been modified accordingly. If the reviewer and / or editor do not agree with the rationale for displaying figure part Figure 1h, left (new Figure 1h), then we are willing to remove it.

PDB files are now provided to the reviewer.

“2. The authors mention they used HNADOCK to generate the structures of the DNA elements. However, they do not explain how they did that. When looking at HNADOCK (<http://huanglab.phys.hust.edu.cn/hnadock/>), I do not see an option to generate double stranded DNA structures. The procedure needs to be explained in detail so that it can be reproduced. It might be that this procedure leads to distortions in the DNA structure. With these distortions, I am not convinced the docking results are reliable even though the results seem to fit very well with the experimental data. Are the authors able to reproduce their own crystal structure if they generate ISRE DNA with the same protocol?”

We thank the reviewer for this comment and the resulting further clarification of the applied procedure. Indeed, for HNADOCK, only structure imports are supported for DNA, whereas both sequence and structure input options are provided for RNA. Therefore, unlike for RNA, the option to generate a DNA structure does not exist in HNADOCK. Thus, we indeed initially generated ssDNA structures in PyMOL v2.5 and then imported and energy minimized these structures in HNADOCK to generate the dsDNA. The Methods section has been modified accordingly.

When the ISRE DNA was generated with the same protocol as described, we were able to produce a remarkably similar structure to that seen in our crystal structure (PDB 7JM4) as demonstrated in the figure provided as Figure 1 for review only. This result further supports the fact that our docking results are indeed reliable. In order to not further increase the complexity of our manuscript, we suggest to provide this figure for review only.

“3. In their modified text, the authors specify: “All the docking studies and modelling were performed using standard HADDOCK 2.2 with flexible docking parameters and without any additional restraints (AIRs).”

This is a very unclear statement. First, the authors write they did not use any restraints, then they specify that HADDOCK is a docking software that uses ambiguous interaction restraints. Does this mean that the authors did not use any of these ambiguous restraints in HADDOCK? If this is the case, can the authors reproduce their previous crystal structure with a such a protocol devoid of any restraints?”

For further clarification, the Methods section has been modified accordingly. More specifically, the order of the two statements has been swapped so that we first describe how the HADDOCK software operates and then state that no additional restraints were used. The revised text now reads: “All docking studies and modelling were performed using standard HADDOCK 2.2. The HADDOCK programme differs from *ab-initio* docking methods by utilising information from known or predicted protein interfaces for ambiguous interaction restraints (AIRs) and utilizes flexible docking. The default HADDOCK program was used without additional restraints.”

“Moreover, they mention “flexible docking” without clearly specifying the protocol. HADDOCK uses different stages of flexible docking. It is not explained which stages the authors used. Was flexible refinement done with MD simulations or only with energy minimization?”

No MD simulations were reported in this study. Flexible docking was performed as a default option in HADDOCK with default energy minimization. The Methods section was modified accordingly.

“How many models were generated? How many clusters? The “cluster size” in Extended Data Table 4 is for which cluster (what does “best or top complex structure” refer to)? What RMSD is shown in the same table (e.g. interface, entire protein, entire complex, which atoms were used for the fit)? All these are questions that are still not answered.”

This information has now been included in the legend of Extended Data Table 4: “Up to 100 models were generated for each of the complexes and these models were classified into 6 to 12 clusters of varying sizes. The number of models that make up the highest-ranked cluster (cluster size) are shown. The RMSD refers to the entire complex.”

“4. It remains unclear what is superposed in Expanded Data Figure 4a (not written in the legend). Are these docked complexes of the WT and C99R mutant on the ISRE DNA? Or is the crystal structure in the case of the WT? The Extended Data Table 4 does not contain data for the docking of the WT to ISRE which suggests the authors did not perform this experiment. However, the orange DNA looks more distorted than the crystal structure.”

This point has now been clarified in the legend to Extended Data Figure 4a. The modified legend to Extended Data Figure 4a now reads: “(a) Superposition of IRF4-WT (X-ray crystal structure (PDB: 7JM4), orange DNA backbone) and IRF4-C99R (structural model, yellow DNA backbone) with ISRE DNA. IRF4-C99R does not bind to intact ISRE DNA (see Fig. 1g) and needs to shift the ISRE DNA for binding to occur (shown here). The unbiased and reference

free structural modelling of IRF4-C99R with the ISRE DNA shows bending of the DNA and displacement of the phosphate backbone (displaced DNA strand marked in yellow) to accommodate the recognition helix into the major groove of the DNA. Without this structural rearrangement of ISRE DNA, binding of IRF4-C99R to ISRE DNA is difficult.”

REVIEWER COMMENTS

Reviewer #2 (Remarks to the Author):

I appreciate the effort the authors spent to try to clarify the structural modeling and docking experiments. Some aspects (especially regarding the procedure) were clarified and I also appreciate that the PDB files were provided. However, major issues with the data remain and because of these I am not able to recommend the inclusion of these data in current form in the paper (neither in the main text, nor in supplementary materials). In fact, after inspecting the PDBs provided I consider that the entire structural modeling and docking procedure need to be redone if the authors wish to have it in the paper. I still consider a structural based docking analysis an elegant way to look at the different binding specificity of the C99R mutant.

The most important issue is that the structure of the model IRF-C99R-AICE2 (PDB file provided "Fig11 AICE2_C99R.pdb") shows unreasonably distortions in the DNA structure. According to the data presented this should be a competent complex, therefore the structure of the DNA should not present any major distortions except of those expected for such a protein-DNA complex (e.g. some bending, some changes in major and minor groove widths). The PDB provided shows the following major distortions: (i) bases C30, T29, G9 are not base paired; G9 from one strand is in between C30 and T29 from the other strand but the H bond donor-acceptor distances are too long for base pairing (ii) basepair A10-T28 does not have an optimal geometry (WC hydrogen bonds not present), (iii) there is a very unusual position of A10 and A26 which lie as if they are forming a base pair; (iv) A26 does not have a planar geometry of the base; (v) R99 is too close to DNA bases (some distances lower than 2Å), (v) the DNA is massively distorted at both ends (beyond acceptable end effects that may occur). All these are unreasonable distortions for a competent protein-DNA complex.

I do understand that distortions in the structures containing non-competent complexes may occur due to binding incompatibility. However, given the unreasonable distortions in the structure of a competent complex (see point above), I consider that all distortions are probably artifacts of the procedure used,

It is now clear that in Figure 1g there were no docking experiments performed but only a simple substitution of C99 with R in the crystal structure. While this is clear in the response to my comments, it is still not clear from the figure legend which still mentions "computational modeling of IRF4-C99R docking" for the panel 1g.

In all my comments that refer to "distorted DNA structures" I do not refer to the visually distorted backbone representation. I understand that this may be an artifact of the rendering representation. However, I should also add that when I try to re-create the same pictures from the crystal structure in a

visualization program (VMD, Pymol, Chimera), using a (very) thin ribbons or trace representation I do not observe the level of visual distortion apparent in the presented figure 1g. For example a thin “NewRibbons” representation with B-Spline interpolation in VMD creates a beautiful smooth thin line through the backbone using the crystal structure. This is certainly not essential for the manuscript, but it does improve the aesthetics of the figures.

RESPONSE TO REVIEWERS' COMMENTS

Response to the comments of Reviewer #2

As outlined below, we have addressed all points raised by the referee. The repeated or paraphrased comments of the reviewer are shown in *italics*. All changes in the manuscript are marked-up in the respective marked-up version of the manuscript (underlined and highlighted in yellow).

Reviewer #2

“I appreciate the effort of the authors spent to clarify the structural modeling and docking experiments. Some aspects (especially regarding the procedure) were clarified and I also appreciate that the PDB files were provided. However, major issues with the data remain and because of these I am not able to recommend inclusion of these data in current form in the paper (neither in the main text, nor in supplementary materials). In fact, after inspecting the PDBs provided I consider that the entire structural modeling and docking procedure need to be redone if the authors wish to have it in the paper. I still consider a structural based docking analysis an elegant way to look at the different binding specificity of the C99R mutant.

The most important issue is that the structure of the model IRF-C99R-AICE2 (PDB file provided “Fig11 AICE2_C99R.pdb”) shows unreasonably distortions in the DNA structure. According to the data presented this should be a competent complex, therefore the structure of the DNA should not present any major distortions except of those expected for such a protein-

DNA complex (e.g. some bending, some changes in major and minor groove widths). The PDB provided shows the following major distortions: (i) bases C30, T29, G9 are not base paired; G9 from one strand is between C30 and T29 from the other strand but the H bond donor-acceptor distances are too long for base pairing (ii) basepair A10-T28 does not have an optimal geometry (WC hydrogen bonds not present), (iii) there is a very unusual position of A10 and A26 which lie as if they are forming a base pair; (iv) A26 does not have a planar geometry of the base; (v) R99 is too close to DNA bases (some distances lower than 2Å), (vi) the DNA is massively distorted at both ends (beyond acceptable end effects that may occur). All these are unreasonable distortions for a competent protein-DNA complex.

I do understand that distortions in the structures containing non-competent complexes may occur due to binding incompatibility. However, given the unreasonable distortions in the structure of a competent complex (see point above), I consider that all distortions are probably artifacts of the procedure used.”

We thank the reviewer for this comment. We see the point of the reviewer regarding the geometry issues of the base pairing. This might be related to the fact that this is a model and not an experimentally driven crystal structure. We would like to point out the same modelling approach results in an absolutely reliable model of IRF4-WT:ISRE, as demonstrated in “Fig. 1 for review only” in the previous round the review process, which demonstrates that our modelling approach results in robust and reliable models.

However, as discussed with the editors and taking into consideration the comments and concerns of the reviewer, we have now moved the AICE1 modelling from Figure 1h to Extended data Figure 4b, and have now discussed the potential limitations of the AICE1 modelling due to DNA distortions within the respective section of the “Results” section of our revised MS.

Also, we acknowledge the potential limitations of the AICE2-modeling data, in particular due to the DNA distortions discussed. After discussing this issue with the editors and reviewer#2, we agree to remove these data from our MS to avoid any confusion and misinterpretations. While these AICE2 docking analyses would have been an elegant way to further look at the different binding specificities by the C99R mutant, they are not definitive and not essential to the main conclusion of the manuscript.

“It is now clear that in Figure 1g there were no docking experiments performed but only a simple substitution of C99 with R in the crystal structure. While this is clear in the response to

my comments, it is still not clear from the figure legend which still mentions “computational modelling of IRF4-C99R docking” for the panel 1g.”

We thank the reviewer for this comment. The legend has been modified accordingly.

“In all my comments that refer to “distorted DNA structures” I do not refer to the visually distorted backbone representation. I understand that this may be an artifact of the rendering representation. However, I should also add that when I try to re-create the same pictures from the crystal structure in a visualization program (VMD, Pymol, Chimera), using a (very) thin ribbons or trace representation I do not observe the level of visual distortion apparent in the presented figure 1g. For example a thin “NewRibbons” representation with B-Spline interpolation in VMD creates a beautiful smooth thin line through the backbone using the crystal structure. This is certainly not essential for the manuscript, but it does improve the aesthetics of the figures.”

We thank the reviewer for this comment. We agree that this might improve the aesthetics of the figures, however we also think that this is not essential to the manuscript and thus we prefer to keep the figure part as it is at the current stage of the MS.